# LOW COMPLEXITY APPROXIMATE BAYESIAN LOGISTIC REGRESSION FOR SPARSE ONLINE LEARNING

## ABSTRACT

Theoretical results show that Bayesian methods can achieve lower bounds on regret for online logistic regression. In practice, however, such techniques may not be feasible especially for very large feature sets. Various approximations that, for huge sparse feature sets, diminish the theoretical advantages, must be used. Often, they apply stochastic gradient methods with hyper-parameters that must be tuned on some surrogate loss, defeating theoretical advantages of Bayesian methods. The surrogate loss, defined to approximate the mixture, requires techniques as Monte Carlo sampling, increasing computations per example. We propose low complexity analytical approximations for sparse online logistic and probit regressions. Unlike variational inference and other methods, our methods use analytical closed forms, substantially lowering computations. Unlike dense solutions, as Gaussian Mixtures, our methods allow for sparse problems with huge feature sets without increasing complexity. With the analytical closed forms, there is also no need for applying stochastic gradient methods on surrogate losses, and for tuning and balancing learning and regularization hyper-parameters. Empirical results top the performance of the more computationally involved methods. Like such methods, our methods still reveal per feature and per example uncertainty measures.

## 1 INTRODUCTION

We consider *online* (Bottou, 1998; Shalev-Shwartz et al., 2011) binary logistic regression over a series of rounds $t \in \{1, 2, \ldots, T\}$. At round $t$, a *sparse* feature vector $\mathbf{x}_t \in [-1, 1]^d$ with $d_t \ll d$ nonzero values, is revealed, and a prediction for the label $y_t \in \{-1, 1\}$ must be generated. The dimension $d$ can be huge (billions), but $d_t$ is usually tens or hundreds. Logistic regression is used in a huge portion of existing learning problems. It can be used to predict medical risk factors, to predict world phenomena, stock market movements, or click-through-rate in online advertising. The online sparse setup is also very common to these application areas, particularly, if predictions need to be streamed in real time as the model keeps updating from newly seen examples.

A prediction algorithm attempts to maximize probabilities of the observed labels. Online methods sequentially learn parameters for the $d$ features. With *stochastic gradient* methods (Bottou, 2010; Duchi et al., 2011), these are weights $w_{i,t}$ associated with feature $i \in \{1, \cdots, d\}$ at round $t$. Bayesian methods keep track of some distribution over the parameters, and assign an expected *mixture* probability to the generated prediction (Hoffman et al., 2010; Opper & Winther, 1998). The overall objective is to maximize a sequence likelihood probability, or to minimize its negative logarithm. A benchmark measure of an algorithm's performance is its *regret*, the excess loss it attains over an algorithm that uses some fixed *comparator* values of $\mathbf{w}^* \overset{\triangle}{=} (w_1, w_2, \ldots, w_d)^{\mathcal{T}}$ ($\mathcal{T}$ denoting transpose). A comparator $\mathbf{w}^*$ that minimizes the cumulative loss can be picked to measure the regret relative to the best possible comparator in some *space of parameter values*.

Kakade & Ng (2005); Foster et al. (2018); Shamir (2020) demonstrated that, *in theory*, Bayesian methods are capable to achieve regret, logarithmic with the horizon $T$ and linear with $d$, that even matches regret lower bounds for $d = o(T)$. Classical stochastic gradient methods are usually implemented as *proper* learning algorithms, that determine $\mathbf{w}_t$ prior to observing $\mathbf{x}_t$, and are inferior in the worst-case (Hazan et al., 2014), although, in many cases depending on the data, they can still achieve logarithmic regret (Bach, 2010; 2014; Bach & Moulines, 2013). Recent work (Jézéquel et al., 2020) demonstrated non-Bayesian improper gradient based algorithms with better regret.

Unfortunately, superiority of Bayesian methods diminishes by their intractability. A theoretically optimal prior has a diagonal covariance matrix, with each component either uniformly or Normally distributed with large variance. Effects of such a prior cannot be maintained in practical online problems with a large sparse feature set, as the posterior of such a prior no longer has the properties of the prior, but must be maintained as a subsequent prior. Gaussian approximations that rely on diagonalization of the covariance must be used. Neither normal nor the diagonal assumptions are true for the real posterior (even with diagonal prior). They thus lead to performance degradations. Diagonalization is similar to linearization in convex optimization, specifically for *stochastic gradient descent* (SGD) (Zinkevich, 2003). It allows handling features independently, but limits performance.

Bayesian learning literature focused on applying such methods to predict posterior probabilities, and provide model (*epistemic*) uncertainty measurements (Bishop, 2006; Dempster, 1968; Huelsenbeck & Ronquist, 2001; Knill & Richards, 1996). However, uncertainty of a feature is, in fact, mostly a function of the number of examples in which it was present; a measure that can be tracked, not estimated. Methods, such as *Variational Bayesian (VB) Inference* (Bishop, 2006; Blei et al., 2017; Drugowitsch, 2019; Drugowitsch et al., 2019; Ranganath et al., 2014), track such measurements by matching the posterior. However, as demonstrated in Rissanen (1984) seminal work, minimizing regret is identical to uncertainty reduction, as regret is merely a different representation of uncertainty. Regret can be *universally* minimized over the possible parameter space through a good choice of a prior. Hence, to minimize uncertainty, the role of an approximation is *to preserve the effect of such a prior at least in the region of the distribution that dominates the ultimate posterior at the horizon $T$*. This is a simpler problem than matching the posterior, and opens possibilities for simpler approximations that can lead to results identical to those of heavy methods as VB.

VB methods are typically used offline to match a tractable posterior to the true one by upper bounding overall loss. They are computationally involved, requiring either iterative techniques (Bishop, 2006; Blei et al., 2017; Murphy, 2012) like Expectation Maximization (EM) (Dempster et al., 1977; Moon, 1996); or Monte Carlo (MC) sampling, replacing analytical expectation by an empirical one over a randomly drawn set. To converge, MC can be combined with gradient descent, either requiring heavy computations, or adapting *stochastic* gradient methods to update posteriors (Broderick et al., 2013; Knowles, 2015; Nguyen et al., 2017a;b). For online problems, the posterior of an example is the prior of the subsequent one. To minimize losses, online VB must converge to the optimal approximation at every example. Otherwise, additional losses may be incurred, as the algorithm may not converge at each example, while the target posterior keeps moving with subsequent examples. Moreover, combining methods that need to tune hyper-parameters defeats the parameter free nature (Abdellaoui, 2000; Mcmahan & Streeter, 2012; Orabona & Pál, 2016) of Bayesian methods.

Most Bayesian literature addressed the *dense* problem, where $\mathbf{x}_t$ consists of mostly nonzero entries for every $t$, and *the dimension $d$ of the feature space is relatively small*. Techniques, like *Gaussian Mixtures* (Herbrich et al., 2003; Montuelle et al., 2013; 2014; Rasmussen, 2000; Reynolds et al., 2000; Sung, 2004), that may use VB, usually apply matrix computations quadratic in $d$ on the covariance matrix. In many practical problems, however, a very small feature subset is present in each example. For *categorical* features, only one of the features in the vector is present at any example. Techniques, useful for the low dimensional dense problem, may thus not be practical.

**Paper Contributions:** We provide a simple analytical Bayesian method for online sparse logistic and probit regressions with closed form updates. We generalize the method also for dense multi-dimensional updates, if the problem is not completely sparse. Our results are first to study regret for Bayesian methods that are simple enough to be applied in practice. They provide an example to the connection between uncertainty and regret, and more broadly the *Minimum Description Length (MDL)* principle (Grunwald, 2007; Rissanen, 1978a;b; 1984; 1986; Shamir, 2015; 2020). Empirical results demonstrate the advantages of our method over computationally involved methods and over other simpler approximations, both by achieving better regret and better loss on real data. As part of the algorithm, uncertainty measures are provided with no added complexity. We specifically demonstrate that it is sufficient to have an approximation focus on the location of the peak of the posterior and its curvature or value, which are most likely to dominate regret, instead of approximating the full posterior, which brings unnecessary complexity missing the real goal of preserving the effects of a good prior. In fact, approximating the full posterior may eventually lead to poor generalization and overfitting by focusing too much on the tails of the posterior. Our approach *directly* approximates the posterior, unlike VB methods that approximate by minimizing an upper bound on the loss. Finally, our approach leverages sparsity to solve a sparse problem.

**Related Work:** The simplest single dimensional online logistic regression problem ($d = 1$ and $x_{1,t} = 1, \forall t$) was widely studied. Jefferys' prior, $\rho(\theta) \triangleq 1/\left(\pi\sqrt{\theta(1-\theta)}\right)$, is asymptotically optimal (Clarke & Barron, 1994; Xie & Barron, 1997; 2000; Drmota & Szpankowski., 2004)). It can be expressed in terms of *log-odds* weights $w$ as $\rho(w) = e^{w/2}/[\pi\left(1 + e^w\right)]$. Applying a mixture leads to the *Krichevsky & Trofimov (1981)* (KT) add-$1/2$ estimator $Q\left(y_t|y^{t-1}\right) = [n_{t-1}\left(y_t\right) + 0.5]/t$, where $n_{t-1}(y_t)$ counts occurrences of $y_t$. We use $y^t$ to express a sequence from 1 to $t$. Applying this prior in a *Follow The Regularized Leader* (FTRL) setting (McMahan, 2011) also leads to the KT estimator. This raised the question whether regret optimality generalizes to large dimensions (McMahan & Streeter, 2012). Hazan et al. (2014) showed that this was not the case for proper methods. The bounds, however, theoretically generalize for Bayesian methods (Kakade & Ng, 2005; Foster et al., 2018; Shamir, 2020), with large variance Gaussian or uniform prior with diagonal covariance. Peaked priors fail, as for each feature in an example, other features provide a *self excluding log-odds prior*, that shifts the relation between the overall distribution and the feature weight. While wide priors are good theoretically, because of the intractability of the Bayesian mixture integrals, diagonal approximations that are used unfortunately degrade their effect.

Bayesian methods have been studied extensively for estimating posteriors and uncertainty (Bishop, 2006; Makowski et al., 2002; Sen & Stoffa, 1996). There is ample literature researching such techniques in deep networks (see, e.g., Blundell et al. (2015); Hwang et al. (2020); Kendall & Gal (2017); Lakshminarayanan et al. (2017); Malinin & Gales (2018); Wilson (2020)). Most of the work focuses on the ultimate posterior after the full training dataset has been visited. One attempts to leverage the uncertainty measurements to aid in inference on unseen examples. Techniques like *expectation propagation (EP)* (Minka, 2001; 2013) (see also Bishop (2006); Chu et al. (2011); Cunningham et al. (2011); Graepel et al. (2010)) and VB are used to generate estimates of the posterior. In a dense setup, where there is a relatively small number features (or units in a deep network), Gaussian Mixture models can also be learned, where a jointly Gaussian posterior is learned, usually with some kernel that is used to reduce the dimensionality of the parameters that are actually being trained. Such methods, however, do not fit the sparse online setup.

Variational methods are derived utilizing Jensen's inequality to upper bound loss of expectation by expectation of the negative logarithm of the product of the prior $\rho(\mathbf{w})$ and data likelihood $P(y^T|\mathbf{x}^T, \mathbf{w})$. Normalizing this joint by the expected label sequence probability gives the posterior $P(\mathbf{w}|\mathbf{x}^T, y^T)$. Then, a posterior $Q(\cdot)$ with a desired form is matched by minimizing the KL divergence $KL(Q||P)$, which decomposes into expectation w.r.t. $Q(\dot)$ over the loss on $y^T$ and $KL(Q||\rho)$ between the approximated posterior and the true prior. The first term may require techniques like the iterative mean field approximation EM (Bishop, 2006; Jaakkola & Jordan, 1998), or MC sampling to be approximated. Gradient methods can also minimize $KL(Q||P)$. In the sparse setup, it is standard to assume a diagonal $Q(\cdot)$. In an online setting, the process can be iterated over the examples (or mini-batches), where the posterior at $t$ is the prior at $t+1$. Computing the approximate posterior may be very expensive if done for every example. SGD can be used with MC sampling, but that would incur additional losses, as the posterior changes between successive examples. Like VB, EP minimizes the opposite divergence $KL(P||Q)$ between the posterior and its approximate.

## 2 PRELIMINARIES

Let $\rho_t(\mathbf{w})$ be the prior on the weights at round $t$, where we start by initializing some $\rho_1(\mathbf{w})$. We will assume that $\rho(\cdot)$ is approximated by a diagonal covariance Gaussian, with means $\mu_{i,t}$ and variances $\sigma_{i,t}^2$ for component $i$ at time $t$. Leveraging results in Kakade & Ng (2005); Foster et al. (2018); Shamir (2020), if we restrict $w_i \in [-B, B]$, a uniform prior over this interval or a normal prior with standard deviation proportional to $B$ can be picked. (To approximate a Dirichlet-$1/2$, 0-mean normal prior with variance $2\pi$ can be used.) Observing sparse $\mathbf{x}_t$, the prediction for $y_t$ is given by

$$p_t \triangleq P(y_t|\mathbf{x}_t) = \int_{\mathbf{w}} p(y_t|\mathbf{x}_t, \mathbf{w})\rho_t(\mathbf{w})d\mathbf{w} \triangleq \int_{\mathbf{w}} p_t(y_t, \mathbf{w}|\mathbf{x}_t)d\mathbf{w}, \tag{1}$$

where for binary logistic regression, the probability of the label given the example and weights is given by the Sigmoid of the label weighted dot product of the example and weights

$$p(y_t|\mathbf{x}_t, \mathbf{w}) \triangleq \frac{1}{1 + \exp\left(-y_t\mathbf{x}_t^T\mathbf{w}\right)} \triangleq \text{Sigma}\left(y_t\mathbf{x}_t^T\mathbf{w}\right). \tag{2}$$

The expected prediction $p_t$ in (1) marginalizes out the weights $\mathbf{w}$ according to the prior $\rho_t(\cdot)$ from the joint probability of $\mathbf{w}$ and $y_t$. The prediction $p_t$ is a function also of all prior pairs sequence $\{\mathbf{x}^{t-1}, y^{t-1}\}$ through the prior $\rho_t(\cdot)$. After observing $y_t$, we try to match a (diagonal) posterior $Q(\cdot)$ to the weights that will equal the next round's prior

$$\rho_{t+1}(\mathbf{w}) \stackrel{\triangle}{=} Q_t(\mathbf{w}) \approx p(\mathbf{w}|\mathbf{x}^t, y^t) = \frac{p(y_t|\mathbf{x}_t, \mathbf{w})\rho_t(\mathbf{w})}{P(y_t|\mathbf{x}_t)} = \frac{p(y_t|\mathbf{x}_t, \mathbf{w})\rho_t(\mathbf{w})}{p_t}. \tag{3}$$

Using $\mathcal{S}_T \stackrel{\triangle}{=} \{\mathbf{x}^T, y^T\}$, the logarithmic loss incurred by approximation $Q(\cdot)$ on the sequence of predictions is $L(\mathcal{S}_T, Q) \stackrel{\triangle}{=} -\sum_{t=1}^T \log p_t$. Let $\mathbf{w}^*$ be some fixed comparator in the parameter values' space. Then, the *regret* of approximation $Q(\cdot)$ relative to comparator $\mathbf{w}^*$ is given by

$$\mathcal{R}(\mathcal{S}_T, Q, \mathbf{w}^*) \stackrel{\triangle}{=} L(\mathcal{S}_T, Q) - L(\mathcal{S}_T, \mathbf{w}^*) = -\sum_{t=1}^T \left[ \log p_t + \log(1 + \exp(-y_t \mathbf{x}_t^T \mathbf{w}^*)) \right]. \tag{4}$$

The regret can measure the excess loss relative to the best possible $\mathbf{w}^*$ comparator, if it is chosen.

## 3 MARGINALIZED BAYESIAN GAUSSIAN APPROXIMATION

In this section, we describe the proposed method. First, the Sigmoid is approximated by a normal Cumulative Distribution Function (CDF). A prediction for the label of the current example is generated shrinking the cumulative mean score as function of the cumulative variance over all features. The main idea for updating feature distributions is marginalizing away all other covariates for each feature in an example at a given round, such that the mean and variance of the feature can be updated to match the location of the peak and either its curvature or value for the true marginalized posterior. In Appendix B, we demonstrate the same approach for *Probit Regression*. It follows the same steps, except that it does not require the initial approximation. Finally, Appendix D.1 shows how similar approximation methodology can be used to apply simple multi-dimensional updates instead of marginalized one, the can be performed when sparsity is limited.

**Gaussian Approximation of a Sigmoid:** The relation between the logistic distribution and the Normal one was well studied in the statistics literature (see, e.g., Bishop (2006); Murphy (2012)). The Sigmoid function in (2) can be viewed as a CDF, which can be approximated by a normal CDF $\Phi(z)$ (The inverse of $\Phi(\cdot)$ is the *Probit* function.) The derivative of the Sigmoid function is the 0-mean *Logistic* Probability Density Function (PDF). Matching the PDFs, we have $e^w/(1+e^w)^2 \approx 1/\sqrt{2\pi\sigma^2} \exp\{-w^2/2\sigma^2\}$. This yields that the Sigmoid function can be approximated by a 0-mean Gaussian CDF with variance $8/\pi$. Using the standard 0-mean normal $\Phi(\cdot)$ function, the argument is scaled by the inverse of the standard deviation $\sqrt{\pi/8}$, giving

$$\text{Sigma}(w) \stackrel{\triangle}{=} \frac{1}{1 + e^{-w}} \approx \Phi\left(\sqrt{\frac{\pi}{8}} \cdot w\right). \tag{5}$$

More details about this approximation are in Appendix A.

**Approximation approach and some notation:** With the diagonal and Gaussian assumptions, for each sparse example (with only $d_t \ll d$ nonzero entries in $\mathbf{x}_t$), we can assume that we have a single normal random variable, whose mean is the $\mathbf{x}_t$ weighted mean, and whose variance is the quadratically weighted sum of variances. Denote the example total weight, mean, and variance by

$$w_t \stackrel{\triangle}{=} \sum_{i=1}^d x_{i,t} w_{i,t}, \quad \mu_t \stackrel{\triangle}{=} \sum_{i=1}^d x_{i,t} \cdot \mu_{i,t}, \quad \sigma_t^2 \stackrel{\triangle}{=} \sum_{i=1}^d x_{i,t}^2 \cdot \sigma_{i,t}^2 \tag{6}$$

(where the diagonalization assumption is important for the simplicity of the approximation of $\sigma_t^2$).

Since we consider a sparse problem, there is benefit to breaking the dependencies between features present in a given example and updating each independently. We can achieve that by marginalizing the prior at $t$ over all other features. Because we assume all features are jointly independent Gaussians, we can break the joint prior into a product of two components; one, the marginal of the feature, and the other the marginal of all other features together, i.e., the *self excluding prior*. To

match the posterior, we then marginalize on the latter, and match a single dimensional posterior for each feature. We define the self excluding prior for feature $i$ at time $t$, its mean and variance as

$$w_{-i,t} = \sum_{j=1}^{d} x_{j,t} w_{j,t} - x_{i,t} w_{i,t} = \sum_{j \neq i} x_{j,t} w_{j,t}; \quad \mu_{-i,t} \triangleq \mu_t - x_{i,t} \mu_{i,t}; \quad \sigma_{-i,t}^2 \triangleq \sigma_t^2 - x_{i,t}^2 \sigma_{i,t}^2. \quad (7)$$

**Prediction:** With the *probit* approximation in (5) and the single dimensional variable $w_t$, we can compute $p_t$ in (1), replacing $p(y_t|\mathbf{x}_t, \mathbf{w})$ in (2) by a normal CDF. Approximating this integral (see, e.g. Murphy (2012), Section 8.4.4.2, and Bishop (2006)) gives

$$p_t \approx \text{Sigma}\left( \frac{y_t \mu_t}{\sqrt{1 + \frac{\pi}{8}\sigma_t^2}} \right). \quad (8)$$

This result demonstrates how the prediction variance shrinks the prediction towards probability $0.5$.

**Marginalization:** Given the diagonalization assumption, the prior at $t$ can be expressed as $\rho_t(\mathbf{w}) = \rho_{i,t}(w_i) \cdot \rho_{-i,t}(w_{-i})$, where $\rho_{-i,t}(\cdot)$ is the prior on the self excluding prior of $w_i$. Hence,

$$p(y_t, \mathbf{w}|\mathbf{x}_t) = p(y_t|\mathbf{x}_t, \mathbf{w})\rho_{i,t}(w_i)\rho_{-i,t}(w_{-i}). \quad (9)$$

Marginalizing on $w_{-i}$ gives

$$p(y_t, w_i|\mathbf{x}_t) = \rho_{i,t}(w_i) \int_{-\infty}^{\infty} p(y_t|\mathbf{x}_t, \mathbf{w})\rho_{-i,t}(w_{-i})dw_{-i} \triangleq \rho_{i,t}(w_i) I_{w_{-i},t}. \quad (10)$$

The inner integral, which marginalizes over $w_{-i}$ with its prior $\rho_{-i,t}(w_{-i})$, can be approximated by

$$
\begin{aligned}
I_{w_{-i},t} &= \int_{-\infty}^{\infty} \frac{1}{\sqrt{2\pi\sigma_{-i,t}^2}} \exp\left( -\frac{(w_{-i} - \mu_{-i,t})^2}{2\sigma_{-i,t}^2} \right) \cdot \text{Sigma}\left[ y_t(x_{i,t}w_i + w_{-i}) \right] dw_{-i} \\
&\overset{(a)}{\approx} \int_{-\infty}^{\infty} \frac{1}{\sqrt{2\pi\sigma_{-i,t}^2}} \exp\left( -\frac{(w_{-i} - \mu_{-i,t})^2}{2\sigma_{-i,t}^2} \right) \cdot \Phi\left[ \sqrt{\frac{\pi}{8}} y_t(x_{i,t}w_i + w_{-i}) \right] dw_{-i} \\
&\overset{(b)}{=} \int_{-\infty}^{\infty} \phi(z) \cdot \Phi\left[ \sqrt{\frac{\pi}{8}} y_t(x_{i,t}w_i + \mu_{-i,t} + \sigma_{-i,t}z) \right] dz \\
&\overset{(c)}{=} \Phi\left( \frac{\sqrt{\frac{\pi}{8}} y_t(\mu_{-i,t} + x_{i,t}w_i)}{\sqrt{1 + \frac{\pi}{8}\sigma_{-i,t}^2}} \right) \\
&\overset{(d)}{\approx} \text{Sigma}\left( \frac{y_t(\mu_{-i,t} + x_{i,t}w_i)}{\sqrt{1 + \frac{\pi}{8}\sigma_{-i,t}^2}} \right). \quad (11)
\end{aligned}
$$

Step $(a)$ follows, again, from the approximation in (5). For $(b)$, we apply the change of variables $z = (w_{-i} - \mu_{-i,t})/\sigma_{-i,t}$, where $\phi(\cdot)$ is the standard Gaussian PDF. The integral in $(b)$ gives $\Phi\left( \frac{a}{\sqrt{1+b^2}} \right)$, with $a = \sqrt{\frac{\pi}{8}} y_t(\mu_{-i,t} + x_{i,t}w_i)$ and $b^2 = \frac{\pi}{8}\sigma_{-i,t}^2$ to lead to $(c)$. Finally, the approximation in (5) is used to go back from a Normal CDF to a Sigmoid in $(d)$.

**Posterior:** The posterior on $w_i$ is given by plugging (11) into (10) normalizing by $p_t$ given in (1).

$$\rho_{i,t+1}(w_i) = Q_{i,t}(w_i) \approx p(w_i|\mathbf{x}^t, y^t) = \frac{1}{p_t} \cdot \rho_{i,t}(w_i) \cdot \text{Sigma}\left( \frac{y_t(\mu_{-i,t} + x_{i,t}w_i)}{\sqrt{1 + \frac{\pi}{8}\sigma_{-i,t}^2}} \right). \quad (12)$$

The approximation on the right implies matching the current true posterior with the $i$th component of the approximate posterior $Q(\cdot)$. It can be simplified to

$$\frac{1}{\sigma_{i,t+1}} \exp\left( -\frac{(w_i - \mu_{i,t+1})^2}{2\sigma_{i,t+1}^2} \right) \approx \frac{1}{p_t \sigma_{i,t}} \exp\left( -\frac{(w_i - \mu_{i,t})^2}{2\sigma_{i,t}^2} \right) \cdot \text{Sigma}\left( \frac{y_t(\mu_{-i,t} + x_{i,t}w_i)}{\sqrt{1 + \frac{\pi}{8}\sigma_{-i,t}^2}} \right). \quad (13)$$

**Approximations:** Because the functional form of the posterior is not Gaussian, there are multiple ways to fit a Gaussian. We review alternatives in Appendix D. However, we want to ensure that the regions of the true posterior we are most likely to converge to at the horizon are not scaled down too much, as this will incur additional loss. It is thus desirable to match the peak of the true posterior with the peak of the approximation. One method is to match both the location and hight of the peak. The other, Laplace approximation (Bishop, 2006), matches the location and curvature at the peak. Both methods give the same approximate for $\mu_{i,t+1}$, but a somewhat different one for $\sigma^2_{i,t+1}$.

To give $\mu_{i,t+1}$, we find $w_i$ that maximizes the r.h.s. of (13), or minimizes its negative logarithm. Let

$$p_{i,t+} \overset{\triangle}{=} \text{Sigma}\left(\frac{y_t\left(\mu_{-i,t} + x_{i,t}\mu_{i,t+1}\right)}{\sqrt{1 + \frac{\pi}{8}\sigma^2_{-i,t}}}\right) = \left[1 + \exp\left(-\frac{y_t\left(\mu_{-i,t} + x_{i,t}\mu_{i,t+1}\right)}{\sqrt{1 + \frac{\pi}{8}\sigma^2_{-i,t}}}\right)\right]^{-1} \quad (14)$$

be almost $p_t$ in (8), except that $\mu_{i,t+1}$ replaces $\mu_{i,t}$ and $\sigma^2_{-i,t}$ replaces $\sigma^2_t$. Thus $p_{i,t+}$ is the probability predicted for $y_t$ if we update $\mu_{i,t}$ and shrink as function of $\sigma^2_{-i,t}$. The minimization gives

$$\mu_{i,t+1} = \mu_{i,t} + \frac{y_t x_{i,t} \sigma^2_{i,t}}{\sqrt{1 + \frac{\pi}{8}\sigma^2_{-i,t}}} \cdot (1 - p_{i,t+}). \quad (15)$$

Eq. (15) can be solved iteratively, where the Newton's method can be used, as described in Appendix C. The solution for $\mu_{i,t+1}$ can also be expressed in terms of the $r$ *generalized Lambert W function* (Corless et al., 1996; Mezo & Baricz, 2015).

Alternatively, to avoid multiple iterations per update when using Newton's method, we can use a Taylor series approximation of $1 - p_{i,t+}$ around $1 - p_{i,t}$, where

$$p_{i,t} \overset{\triangle}{=} \text{Sigma}\left(\frac{y_t\left(\mu_{-i,t} + x_{i,t}\mu_{i,t}\right)}{\sqrt{1 + \frac{\pi}{8}\sigma^2_{-i,t}}}\right) = \text{Sigma}\left(\frac{y_t\mu_t}{\sqrt{1 + \frac{\pi}{8}\sigma^2_{-i,t}}}\right). \quad (16)$$

Like $p_{i,t+}$, $p_{i,t}$ is not $p_t$. Instead, it is the probability of $y_t$ as projected by the means of the weights at $t$, shrunk as function of $\sigma^2_{-i,t}$ instead of $\sigma^2_t$. More importantly, it depens only on parameters before the update at $t + 1$ is applied, giving a closed form solution. Applying first order approximation we have

$$\mu_{i,t+1} = \mu_{i,t} + \frac{y_t x_{i,t} \sigma^2_{i,t}\left(1 - p_{i,t}\right)}{\sqrt{1 + \frac{\pi}{8}\sigma^2_{-i,t}}\left[1 + \frac{1}{1+\frac{\pi}{8}\sigma^2_{-i,t}}y_t^2 x_{i,t}^2 \sigma^2_{i,t}\left(1 - p_{i,t}\right)p_{i,t}\right]}. \quad (17)$$

If may be simpler to store the *precision* $1/\sigma^2_{i,t}$, in which case, (17) may be easier to compute by normalizing both numerator and denominator by $\sigma^2_{i,t}$, applying this normalization on the right term of the denominator. Second or higher orders approximations can also be applied, but may not be necessary, as the first order one already gives identical performance to the iterative method.

After updating $\mu_{i,t+1}$, we can apply (13) to update $\sigma_{i,t+1}$. Plugging (15) in (13), we solve for $\sigma_{i,t+1}$,

$$\sigma_{i,t+1} = \frac{p_t\sigma_{i,t}}{p_{i,t+}}\cdot\exp\left\{\frac{(\mu_{i,t+1} - \mu_{i,t})^2}{2\sigma^2_{i,t}}\right\} = \frac{p_t\sigma_{i,t}}{p_{i,t+}}\cdot\exp\left\{\frac{y_t^2 x_{i,t}^2 \sigma^2_{i,t}}{2(1 + \frac{\pi}{8}\sigma^2_{-i,t})}\cdot(1 - p_{i,t+})^2\right\}. \quad (18)$$

Alternatively to (18), Laplace approximation can be used by finding the second derivative of the negative logarithm of the posterior, giving

$$\sigma^2_{i,t+1} = \left[\frac{1}{\sigma^2_{i,t}} + \frac{y_t^2 x_{i,t}^2}{1 + \frac{\pi}{8}\sigma^2_{-i,t}}\cdot p_{i,t+}\cdot(1 - p_{i,t+})\right]^{-1}. \quad (19)$$

The procedures described are summarized in Algorithm 1. In Appendix D, we describe several different methods and approximations that can be used, including one that applies the same approximation steps we applied in this section without the marginalization. As empirical results, in the next section, show, however, there is no performance advantage to applying any of the more involved methods.

---

**Algorithm 1** Marginalized Bayesian Gaussian Approximation

1: **procedure** MARGINALIZED BAYESIAN GAUSSIAN APPROXIMATION(Parameters: $\mu_0$, $\sigma_0^2$)
2:      $\forall i \in 1, \ldots, d; \mu_{i,1} \leftarrow \mu_0, \sigma_{i,1}^2 \leftarrow \sigma_0^2$.
3:     **for** t=1,2,...,T **do**
4:        Get $\mathbf{x}_t$.
5:        Compute $\mu_t$, $\sigma_t^2$ with (6).
6:        Generate $p_t$ for $y_t \in \{-1, 1\}$ with (8).
7:        Observe $y_t$.
8:        **for** $i : x_{i,t} \neq 0$ **do**
9:           Compute $p_{i,t}$ $p_{i,t+}$ with (16) and (14), respectively, using $\mu_{i,t+1} = \mu_{i,t}$ for (14).
10:        Iterate on (15) and (14) with Newton's method, or use (17) to update $\mu_{i,t+1}$.
11:         Update $\sigma_{i,t+1}^2$ with either (18) or (19)
12:        **end for**
13:     **end for**
14: **end procedure**

---

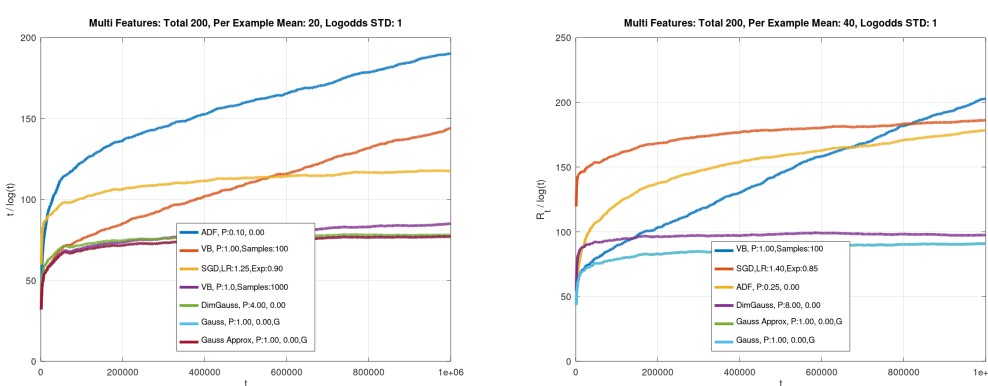

Figure 1: $\mathcal{R}_t / \log t$ vs. round $t$ for different methods for randomly drawn $d$ binary features, expected $d_t/d$ features per example, and standard deviation of true log-odds noted in each graph.

## 4   NUMERICAL RESULTS

To benchmark regret of an algorithm, one needs the ground truth of real or loss minimizing parameters. On real data, the loss minimizing parameters are unknown. Furthermore, true benchmark datasets can consist of non-stationary data, and the feature sets selected by a model one trains may misspecify the "true" features of such a model. Thus real data may not give clean evaluation of the proposed methods. We present results on a benchmark dataset at the end of this section, but to measure regret performance of different algorithms, we present results on synthetic data.

**Synthetic Data:** We simulated data by setting $d$ features with true log-odds weights that were drawn randomly with some prior (or with multiple priors, each governing a subset of the features). At $t$, $d_t \ll d$ features were selected in random from the set of $d$ features, where different rules were used for different simulations to draw the $d_t$ features. In the fully random case, we set a random fraction $\alpha = d_t/d$ parameter, and each feature was activated with probability $\alpha$. We either used binary features $x_{i,t} \in \{0, 1\}$, or also drew $x_{i,t}$ randomly in $[0, 1]$. For categorical features, the $d$ features were partitioned into categories, and for every example, one or a preset number of features from each category were randomly selected, with different types of randomness, including fast decaying long tail distributions. Let $\boldsymbol{\theta}$ be the vector of true parameters, then, $\Pr(Y_t = 1)$ was computed as $\text{Sigma}(\mathbf{x}_t^T \boldsymbol{\theta})$. The probability was used to randomly draw $y_t$. We used Algorithm 1, as well as other algorithms, described in Appendix D, to sequentially predict $y_t$ and update. For gradient methods, we used *Stochastic Gradient Descent (SGD)* with *AdaGrad* scheduling (Duchi et al., 2011).

We ran grids of algorithm hyper-parameters for all algorithms to find optimal ones, and we show results for these optimal hyper-parameters for all algorithms. Since we know the true weights, we

Table 1: Runtime and regret coefficients $r_T = \mathcal{R}_T / \log T$ for different algorithms on synthetic 200 features models with true log-odds std of 1 and algorithm parameters as in Fig 1.

| | Per Example Mean: 20 | | | | Per Example Mean: 40 | | | |
| | 1M Examples | | 10M Examples | | 1M Examples | | 10M Examples | |
| Alg. | Time | $r_T$ | Time | $r_T$ | Time | $r_T$ | Time | $r_T$ |
|---|---|---|---|---|---|---|---|---|
| SGD | 8s | 117.6 | 58s | 135.9 | 10s | 246.3 | 1:18m | 219.88 |
| ADF | 6s | 190.3 | 45s | 540.3 | 7s | 178.6 | 53s | 388.51 |
| DimGauss | 9s | 78.02 | 1:01m | **82.54** | 14s | 97.52 | 1:42m | **109.16** |
| Gauss | 9s | **77.66** | 1:05m | 93.85 | 14s | **91.5** | 1:32m | 144.28 |
| ApproxG | 9s | **77.66** | 1:05m | 93.85 | 12s | **91.5** | 1:32m | 144.28 |
| VB-100 | **3:47m** | 144.22 | **35:25m** | 649.57 | **7:21m** | 202.9 | **1:10:25h** | 1125.3 |
| VB-1000 | **35:16m** | 85.07 | | | **1:11:01h** | 104.31 | | |

use them for a comparator baseline $\mathbf{w}^*$. Curves show *progressive validation* regret. At round $t$ we measure the cumulative regret up to $t$ given by $\mathcal{R}_t \triangleq - \sum_{\tau=1}^{t} \left[ \log p_\tau + \log(1 + \exp(-y_\tau \mathbf{x}_\tau^T \mathbf{w}^*)) \right]$. Instead of showing $\mathcal{R}_t$, we plot $\mathcal{R}_t / \log t$. If an algorithm has logarithmic regret, the normalized curve of the algorithm will converge to the constant. This methodology thus allows us to observe whether an algorithm has logarithmic regret or not. Results are shown for Algorithm 1 with its different variations (labeled by *Gauss* for updates with (15), and by *Gauss Approx* with (17)). Labels also designate the prior used ($\sigma_0^2$ and $\mu_0$), and which variance approximation was used (G for (18) and L for (19)). Reference results are shown for SGD, multi-dimensional Gaussian approximation update (DimGauss) described in Appendix D.1, EP; using *Assumed Density Filtering (ADF)*, and marginalized VB (VBApprox), described in Appendix D.3.

Fig. 1 shows normalized regret for two different true data configurations described at the top of each graph with random binary features. More detailed results on multiple configurations are shown in Appendix E. Unfortunately, unlike theoretical results (Shamir, 2020), the prior has to fit the data for all methods of Gaussian approximation for good regret. This is true for *any* known practical Bayesian method, as well as for the learning parameters of SGD. However, if we choose a prior that matches the true prior, regret rates logarithmic in $T$, usually close to the lower bounds of $0.5 \log T$ per parameter, are achieved on all experiments with Algorithm 1. The same results are achieved whether we use the iterative version in (15) or its simpler approximation (17), and whether the variance is updated by (18) or (19). Unlike Algorithm 1, both, DimGauss and ADF appear to be optimized for priors that are different from the true one, and that depend on the fraction or number of features that occur in each example. DimGauss requires larger prior with more features. As the sparsity is reduced, DimGauss with its optimal prior seems to improve relative to Algorithm 1. This is expected, as the sparsity assumption becomes less valid. Algorithm 1 outperforms both ADF and SGD in all cases. We can find SGD hyper-parameters that seem to still exhibit logarithmic regret for each configuration, but are inferior to Algorithm 1, with increasing gaps with more active features.

Table 1 shows both execution runtimes and regret coefficients $r_T \triangleq \mathcal{R}_T / \log T$ for the full simulations with the algorithms and configurations in Fig 1. Simulations were run on a single Ubuntu machine and included synthetic data generation and similar outputs for all algorithms compared. DimGauss, VB, and ADF may have a slight advantage, as they were implemented with the Eigen package, which is highly optimized for matrix operations. We show benchmarks for $10^6$ and $10^7$ examples. We observe rather equal runtimes for SGD, ADF, DimGauss and Algorithm 1, with slight advantage to ADF, which may be due to the highly optimized matrix operations. Good regret is obtained for both methods of Algorithm 1, but DimGauss slowly improves over Algorithm 1. This is true because selecting 20 or 40 features out of 200 results in repetitions of co-occurrences as more data is observed, and DimGauss utilizes these co-occurrences. We show below and also observe from the results in Appendix E, however, that with higher degrees of sparsity, the DimGauss algorithm degrades, while Algorithm 1 retains its advantages. Interestingly, the Newton method in the Gauss algorithm does not require more time than its Taylor approximation, and both regret and runtime are matched between the two. This is attributed to the fact that the Newton method requires very few iterations to converge. Boldfaced in the table are the poor runtime results of the VB meth-

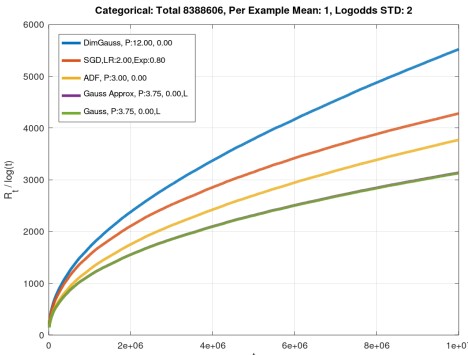 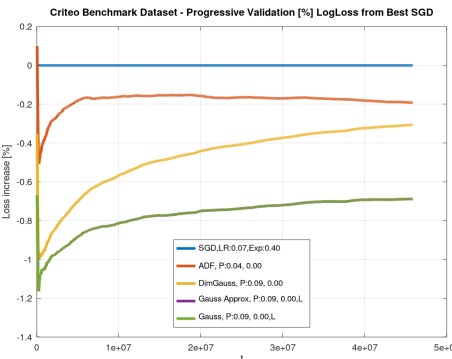

Figure 2: Left: $\mathcal{R}_t / \log t$ vs. $t$ for synthetic data with millions of long tail features. Right: Loss relative to the best SGD on the Criteo benchmark dataset for multiple algorithms.

ods (shown with 100 and 1000 samples), whose worst-case complexity is $O(d_t N J T)$, where $N$ is the number of samples and $J$ is the maximal number of iterations. The table illustrates how runtime increases substantially relative to all other algorithms whose complexity is $O(d_t T)$. Regret only approaches that of Algorithm 1 with 1000 samples, and is far inferior with 100 samples.

Fig. 2 (left) shows normalized regret of the different methods for a synthetic categorical model, with 22 categories. Category $j, j = 1, 2, \ldots$ consists of $2^j$ features (a total of over $8M$). For each example, a single feature is randomly drawn from each category with Zipf distribution $c/(n+1)^{1.75}$, where $c$ is a normalizer and $n$ is the feature index in its category. The true weights are randomly drawn as before, with standard deviation 2. This gives a long tail distribution over features, where for each category small indices are drawn more often, but many features from the long tail do occur in examples. This models a realistic sparse dataset. Algorithm 1 outperforms other methods, and the best values of the DimGauss algorithm are inferior due to the sparsity.

**Criteo display advertising challenge benchmark Dataset:** The right graph in Fig. 2 gives relative percent aggregate loss performance of the Bayesian algorithms relative to the best configuration we found for an AdaGrad SGD on the Criteo dataset[1]. We trained all algorithms on the over $45M$ examples in this dataset, which consists of 13 integer valued features, and 26 categorial features with different category counts. For each example, we generated a prediction, computed its log loss on the label, and applied update. The aggregate log loss is a sum of data uncertainty and regret. The first term is equal for all algorithms and linear in the size of the data, where the second is the regret, which is sub-linear in the data size for a good algorithm. Thus even small noticeable percent improvements imply possible substantial improvements of regret. Algorithm 1 achieved with *only linear models* $0.465$ progressive validation log loss, which is better, for example, than results reported also using deep networks in (Cheng et al., 2016). We observe advantages to the Bayesian methods over SGD, where Algorithm 1 was superior to all methods. The DimGauss method slowly degrades relative to the other methods due to the sparsity of some of the categorical features.

## 5 CONCLUSIONS

We introduced a simple Bayesian mixture diagonal Gaussian approximation method based on marginalization for sparse online logistic regression and probit regression, that attempts to retain the affects of a good prior around the optimal values of the weights. The method does not require the complexities of standard Bayesian methods, as VB, but was empirically shown to achieve regret rates as good and even better. With proper priors, empirical results were close to regret lower bounds, and superior to other Bayesian methods also measured with their best choices of priors. With the strong relation between regret and uncertainty, this approach gives good uncertainty estimates. The methodology was proposed for logistic regression, and extended for probit regression, but can be further extended to other settings. We also demonstrated a matching approach that performs high dimensional updates, and can be used for dense problems.

---

[1] https://www.kaggle.com/c/criteo-display-ad-challenge

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

## A  RELATION BETWEEN GAUSSIAN AND SIGMOID

The Sigmoid function, which converts log-odds to probability is very close in shape to the Gaussian Cumulative Distribution Function (CDF) $\Phi(z)$, as well established in the statistics literature (see, e.g., Bishop (2006); Murphy (2012)). The derivative of the Sigmoid function is given by

$$\frac{d\mathrm{Sigma}(w)}{dw} = \frac{e^w}{(1+e^w)^2} \tag{20}$$

and equals the PDF of a $0$-mean *Logistic* distribution. We can approximate the logistic PDF by a Gaussian by matching the PDFs,

$$\frac{e^w}{(1+e^w)^2} \approx \frac{1}{\sqrt{2\pi}\sigma} \exp\left\{-\frac{w^2}{2\sigma^2}\right\}. \tag{21}$$

Matching the distributions at $w = 0$ yields $\sigma = \sqrt{8/\pi}$,

$$\frac{e^w}{(1+e^w)^2} \approx \frac{1}{4} \exp\left\{-\frac{\pi w^2}{16}\right\} = \sqrt{\frac{\pi}{8}} \cdot \frac{1}{\sqrt{2\pi}} \exp\left\{-\frac{w^2}{2\cdot\frac{8}{\pi}}\right\}. \tag{22}$$

Thus, we can approximate the Sigmoid with a $0$-mean Gaussian CDF with variance $8/\pi$, giving (5).

It remains to demonstrate that the PDFs (and CDFs) are close to each other not only at the peak. Figure 3 demonstrates the approximation of the Logistic distribution (left) and the Sigmoid function (middle) by a normal PDF and a normal CDF both with variance $8/\pi$, respectively. The green curve shows the differences between the logistic/Sigmoid and the normal, which are also plotted on the right plot at larger scale. The magnitude of the difference between the Sigmoid and the normal CDF is bounded by $0.02$ over the whole region. The differences appear asymmetric around the origin, and are substantially small at $1/3$ standard deviation from the origin, or less. While they can accumulate over multiple examples, it appears that the most probable scenario is that positive and negative differences over multiple examples cancel each other. Furthermore, the motivation of a Bayesian method is to converge toward a peaked point mass, at the loss minimizing value of the parameter. As the variance is narrowed closer to such a point mass, the approximation tends to exist in the flat region around the origin, where the difference between the Sigmoid and the normal CDF is very small.

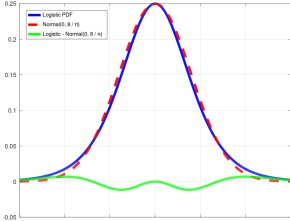 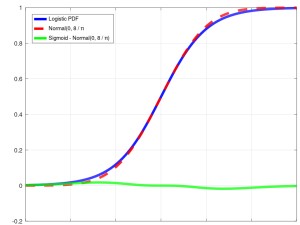 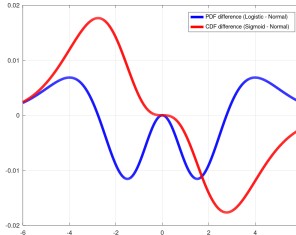

Figure 3: Left: Logistic and Normal $\mathcal{N}(0, 8/\pi)$ distributions and their differences. Middle: Sigmoid and Normal CDF $\mathcal{N}(0, 8/\pi)$ and their differences. Right: Differences between logistic and normal (PDFs and CDFs).

## B PROBIT REGRESSION

In this appendix, we show the derivation of the method proposed in this paper for *Probit Regression*, where, in a similar manner to (2), the predicted label probability with weight vector $\mathbf{w}$, label $y_t$, and covariates $\mathbf{x}_t$ is given by the normal CDF

$$p(y_t|\mathbf{x}_t, \mathbf{w}) \triangleq \int_{-\infty}^{y_t \mathbf{x}_t^T \mathbf{w}} \frac{1}{\sqrt{2\pi}} \exp\left(-\frac{\alpha^2}{2}\right) d\alpha \triangleq \int_{-\infty}^{y_t \mathbf{x}_t^T \mathbf{w}} \phi(\alpha) d\alpha \triangleq \Phi\left(y_t \mathbf{x}_t^T \mathbf{w}\right) \quad (23)$$

where, as we recall, $\phi(\cdot)$ and $\Phi(\cdot)$ are the standard Gaussian (normal) PDF and CDF, respectively. While for logistic regression, we used a Gaussian approximation to obtain analytical expressions for the prediction in (8) and the marginalization integral in (11), for probit regression, these are no longer approximations. For the posterior, we will still apply a Gaussian and a diagonal approximations, as in the derivations based on (13).

**Prediction:** The approach for probit regression is similar to the one described in Section 3 for logistic regression. For each feature we track the mean $\mu_{i,t}$ and the variance $\sigma_{i,t}^2$ for the $i$th feature. For example $t$, we use (6) to compute the total weight $w_t$, its mean $\mu_t$ and variance $\sigma_t^2$. Eq. (7) gives the self excluding weights, their means, and their variances. Similarly to (8), using the approximate normal prior at $t$, we can derive the label prediction for $y_t$,

$$
\begin{aligned}
p_t = P(y_t|\mathbf{x}_t) &= \int_{-\infty}^{\infty} \frac{1}{\sqrt{2\pi\sigma_t^2}} \cdot \exp\left\{-\frac{(w_t - \mu_t)^2}{2\sigma_t^2}\right\} \cdot \Phi\left(y_t \mathbf{x}_t^T \mathbf{w}\right) \cdot dw_t \\
&\overset{(a)}{=} \int_{-\infty}^{\infty} \frac{1}{\sqrt{2\pi\sigma_t^2}} \cdot \exp\left\{-\frac{(w_t - \mu_t)^2}{2\sigma_t^2}\right\} \cdot \int_{-\infty}^{y_t w_t} \frac{1}{\sqrt{2\pi}} \exp\left(-\frac{z^2}{2}\right) \cdot dz \cdot dw_t \\
&\overset{(b)}{=} \int_{-\infty}^{\infty} \frac{1}{\sqrt{2\pi}} \cdot \exp\left\{-\frac{v^2}{2}\right\} \cdot \int_{-\infty}^{y_t(\sigma_t v + \mu_t)} \frac{1}{\sqrt{2\pi}} \exp\left(-\frac{z^2}{2}\right) \cdot dz \cdot dv \\
&\overset{(c)}{=} \int_{-\infty}^{\infty} \phi(v) \cdot \Phi\left[y_t(\sigma_t v + \mu_t)\right] dv \\
&\overset{(d)}{=} \Phi\left(\frac{y_t \mu_t}{\sqrt{1 + \sigma_t^2}}\right).
\end{aligned}
\quad (24)
$$

For $(a)$, we use the definition of $w_t$ in (6). Step $(b)$ follows from substituting $v = (w_t - \mu_t)/\sigma_t$. Step $(c)$ identified the integrands as a product of the standardized $\mathcal{N}(0,1)$ normal PDF multiplied by a standardized normal CDF at $y_t(\sigma_t v + \mu_t)$. This integral gives a normal CDF $\Phi\left(\frac{a}{\sqrt{1+b^2}}\right)$ for $a = y_t \mu_t$ and $b^2 = y_t^2 \sigma_t^2 = \sigma_t^2$ leading to $(d)$.

**Marginalization:** Following the marginalization steps in Section 3, we can express the joint probability of weight $w_i$ and label $y_t$ conditioned on the covariates $\mathbf{x}_t$ and marginalized over all the other

nonzero covariates at example $t$ as in (10) by

$$p(y_t, w_i | \mathbf{x}_t) = \rho_{i,t}(w_i) \cdot \Phi \left( \frac{y_t(\mu_{-i,t} + x_{i,t}w_i)}{\sqrt{1 + \sigma_{-i,t}^2}} \right) \tag{25}$$

where we use the steps of (11), excluding the approximations, to derive (25).

**Posterior:** The posterior on $w_i$ is given as in (12), normalizing $p(y_t, w_i | \mathbf{x}_t)$ by $p_t$ from (24).

$$\rho_{i,t+1}(w_i) \approx p(w_i | \mathbf{x}^t, y^t) = \frac{1}{p_t} \cdot \rho_{i,t}(w_i) \cdot \Phi \left( \frac{y_t(\mu_{-i,t} + x_{i,t}w_i)}{\sqrt{1 + \sigma_{-i,t}^2}} \right). \tag{26}$$

This posterior can now be matched by a normal posterior $Q_{i,t}(w_i)$ as in (13).

**Approximation:** Here, we follow the Laplace approximation applied in Section 3. All other methods mentioned in the paper are also possible. To find $\mu_{i,t+1}$ that minimizes the negative logarithm of the r.h.s. of (26), define, similarly to (14),

$$z_{i,t+} \triangleq \frac{y_t(\mu_{-i,t} + x_{i,t}\mu_{i,t+1})}{\sqrt{1 + \sigma_{-i,t}^2}} \tag{27}$$

as the probit score, which serves as the argument of the normal CDF, where the $i$th mean has been updated, but all other means have not. We can now express the update of the $i$th mean by

$$\mu_{i,t+1} = \mu_{i,t} + \frac{y_t x_{i,t}\sigma_{i,t}^2}{\sqrt{1 + \sigma_{-i,t}^2}} \cdot \frac{\phi(z_{i,t+})}{\Phi(z_{i,t+})}. \tag{28}$$

Eq. (28) is a similar update for probit regression to that of (15) for logistic regression, where the ratio $\phi(z_{i,t+})/\Phi(z_{i,t+})$ replaces $1 - p_{i,t+}$ (and the scaling of the self excluding variance is unnecessary).

As (15), (28) must be solved iteratively because the term $\phi(z_{i,t+})/\Phi(z_{i,t+})$ is a function of $\mu_{i,t+1}$ through the definition of $z_{i,t+}$. As in Section 3, we can use a first order Taylor approximation of $\phi(z_{i,t+})/\Phi(z_{i,t+})$ around its value for $\mu_{i,t}$. Similarly to (16), we define

$$z_{i,t} \triangleq \frac{y_t(\mu_{-i,t} + x_{i,t}\mu_{i,t})}{\sqrt{1 + \sigma_{-i,t}^2}} \tag{29}$$

which is the score before update of all means $\mu_{i,t}$, but unlike the one used to compute $p_t$, normalized by the $i$th self excluding variance $\sigma_{-i,t}^2$ instead of $\sigma_t^2$. With some algebra, this gives a single operation update, similar to that in (17), given by

$$\mu_{i,t+1} = \mu_{i,t} + \frac{y_t x_{i,t}\sigma_{i,t}^2 \phi(z_{i,t})/\Phi(z_{i,t})}{\sqrt{1 + \sigma_{-i,t}^2} \left\{ 1 + \frac{1}{1+\sigma_{-i,t}^2} y_t^2 x_{i,t}^2 \sigma_{i,t}^2 \cdot \frac{\phi(z_{i,t})}{\Phi(z_{i,t})} \cdot \left[ z_{i,t} + \frac{\phi(z_{i,t})}{\Phi(z_{i,t})} \right] \right\}}. \tag{30}$$

The term $z_{i,t} + \phi(z_{i,t})/\Phi(z_{i,t})$ in the denominator replaces $p_{i,t}$ in the logistic regression update equation.

Taking the second derivative of the negative logarithm of the posterior and approximating $1/\sigma_{i,t+1}^2$ by it, gives a single operation update of the variance, similarly to (19),

$$\sigma_{i,t+1}^2 = \left\{ \frac{1}{\sigma_{i,t}^2} + \frac{y_t^2 x_{i,t}^2}{1 + \sigma_{-i,t}^2} \cdot \frac{\phi(z_{i,t+})}{\Phi(z_{i,t+})} \cdot \left[ z_{i,t+} + \frac{\phi(z_{i,t+})}{\Phi(z_{i,t+})} \right] \right\}^{-1}. \tag{31}$$

## C   Mean and Variance Updates

Eq. (15) gives an update for the mean $\mu_{i,t+1}$ that cannot be solved in closed form. This is beacuse $p_{i,t+}$ is a function of $\mu_{i,t+1}$. However, the update is easily solvable with a few iterations of Newton's

method. We start by plugging $\mu_{i,t+1}^{(\ell=0)} = \mu_{i,t}$ for iteration $\ell = 0$. We follow by computing $p_{i,t+}^{(0)}$ with (14). The solution for $\mu_{i,t+1}$ is the value of $w_i$ that minimizes the negative logarithm of the r.h.s. of (12) where $\rho_{i,t}(w_i)$ is Gaussian with mean $\mu_{i,t}$ and variance $\sigma_{i,t}^2$. At iteration $\ell$, we can compute $p_{i,t+}^{(\ell-1)}$ with (14), using $\mu_{i,t+1}^{(\ell-1)}$. The gradient of the negative logarithm of the posterior w.r.t. $w_i$ at $w_i = \mu_{i,t+1}^{(\ell-1)}$ is given by

$$g_\ell = \frac{\mu_{i,t+1}^{(\ell-1)} - \mu_{i,t}}{\sigma_{i,t}^2} - \frac{y_t x_{i,t}}{\sqrt{1 + \frac{\pi}{8}\sigma_{-i,t}^2}} \cdot \left(1 - p_{i,t+}^{(\ell-1)}\right). \tag{32}$$

The second derivative w.r.t. $w_i$ is given by

$$h_\ell = \frac{1}{\sigma_{i,t}^2} + \frac{y_t^2 x_{i,t}^2}{1 + \frac{\pi}{8}\sigma_{-i,t}^2} \cdot p_{i,t+}^{(\ell-1)} \cdot \left(1 - p_{i,t+}^{(\ell-1)}\right) \tag{33}$$

Then, $\mu_{i,t+1}$ is updated by

$$\mu_{i,t+1}^{(\ell)} = \mu_{i,t+1}^{(\ell-1)} - \frac{g_\ell}{h_\ell}. \tag{34}$$

The process terminates when the difference $\left|\mu_{i,t+1}^{(\ell)} - \mu_{i,t+1}^{(\ell-1)}\right|$ is smaller than some threshold, or a specified number of iterations had already executed. Note that the second derivative in (33) (the Newton Hessian step) gives the same expression as the update to the precision (inverse variance) in (19).

Eq. (18) gives an update to the variance. It is interesting to interpret some observations from this update. For $x_{i,t} \geq 0$, $p_{i,t+} \geq p_t$. This leads to $y_t x_{i,t}(\mu_{i,t+1} - \mu_{i,t}) > 0$ because for every $y_t$, $\mu_{i,t+1}$ has to move away from $\mu_{i,t}$ with the sign of $y_t$. This implies that we add a positive term from the log-odds converted to $p_t$ to those converted into $p_{i,t+}$ and also apply less shrinkage using $\sigma_{-i,t}^2$ instead of $\sigma_t^2$, yielding this claim. Hence, the coefficient of $\sigma_{i,t}$ outside the exponential term in (18) is upper bounded by 1. The argument of the exponential term is always nonnegative, which implies that the exponential term is lower bounded by 1. However, if the self-excluding variance $\sigma_{-i,t}^2$ is large, it makes the argument small. Similarly, if $p_{i,t+}$ is large relative to $p_t$ it makes the whole expression smaller. These two observations imply that if the self excluding prior is less certain, or if the self excluding prior has a different belief about the label from the one observed overall, the uncertainty of the current feature reduces more, because it is deemed responsible for the observation $y_t$. On the other hand, if the opposite holds, i.e., either the uncertainty of the self excluding prior is low, or the self excluding prior agrees more with the observed label $y_t$, then, feature $i$ matters less for the observation, and therefore, its uncertainty is not reduced as much.

## D    OTHER METHODS

In this appendix, we describe updates we perform with other Gaussian approximation methods. We can update the mean vector and covariance of $d_t$ features that occur at $t$, without marginalization, discarding off-diagonal covariance terms. This approach, that otherwise uses similar approximations to the ones we used in Section 3, is described in Section D.1, where some of the approximation steps are novel to this paper.

Alternatively, one can choose any two points on the true posterior and match $Q(\cdot)$ on these. Least squares can be used with several points that represent a region of $w_i$, for which the posterior at $T$ is likely to have most mass. EP and VB can be applied, discussed in Sections D.2 and D.3, respectively. For the latter, specifically, single dimensional VB on the marginalized posterior can be applied to minimize $KL(Q_{i,t}(w_i)||p(w_i|\mathbf{x}^t, y^t))$, where $p(w_i|\mathbf{x}^t, y^t)$ is the true posterior on the r.h.s. of (12) given by $p(y_t, w_i|\mathbf{x}_t)/p_t$, where $p(y_t, w_i|\mathbf{x}_t)$ is in (10). This still requires expensive Monte Carlo estimation of expectations with an iterative Newton method. Without marginalization, however, a similar VB approach would require even more expensive Monte Carlo sampling or iterative mean field approximation EM to converge on all components.

### D.1 MULTI-DIMENSIONAL GAUSSIAN APPROXIMATION

Instead of marginalizing on all other features to update $w_i$ for which $x_{i,t} \neq 0$, we can apply multidimensional update on all features for which $x_{i,t} \neq 0$ at round $t$. Such updates will enhance correlation between these features, and may be a better fit to problems in which such correlation is expected. For this update, we assume that the true posterior consists of a product between a prior with a diagonal covariance matrix and a Sigmoid, and we apply Lapace approximation to obtain new mean vector and covariance. With some abuse of notation, let all values at $t$ consist only of the $d_t$ nonzero components of $\mathbf{x}_t$. Let $\Sigma_t$ be the diagonal covariance matrix, with diagonal elements $\sigma_{i,t}^2$. Let $\mathbf{u}_t$ be the estimated mean vector at $t$. Then, the true posterior at $t$ is given by

$$p(\mathbf{w}|\mathbf{x}^t, y^t) = \frac{1}{p_t} \cdot \frac{\exp\left\{-\frac{1}{2}(\mathbf{w} - \mathbf{u}_t)^\mathcal{T} \Sigma_t^{-1}(\mathbf{w} - \mathbf{u}_t)\right\}}{\sqrt{(2\pi)^{d_t} |\Sigma_t|}} \cdot \frac{1}{1 + \exp\left\{-y_t \mathbf{x}_t^\mathcal{T} \mathbf{w}\right\}} \tag{35}$$

Similarly to (14), define

$$p_{t+} \stackrel{\triangle}{=} \text{Sigma}\left(y_t \mathbf{x}_t^\mathcal{T} \mathbf{u}_{t+1}\right) \tag{36}$$

as the probability of $y_t$ computed with weights after they had been updated (and this time with no shrinkage), where $\mathbf{u}_{t+1}$ is the updated vector of means. Then, with Laplace approximation, taking the value of the mean vector that maximizes the posterior, the mean can be updated as in (15) by

$$\mathbf{u}_{t+1} = \mathbf{u}_t + \Sigma_t y_t \mathbf{x}_t \left(1 - p_{t+}\right). \tag{37}$$

This is, again, an equation that must be solved either numerically, or using methods such as Newton's method. Again, we can assign $\mathbf{u}_{t+1}^{(0)} = \mathbf{u}_t$, and apply (36) on it to obtain $p_{t+}^{(0)}$. Then, at iteration $\ell$,

$$\mathbf{g}_\ell = \Sigma_t^{-1}(\mathbf{u}_{t+1}^{(\ell-1)} - \mathbf{u}_t) - y_t \mathbf{x}_t \cdot (1 - p_{t+}^{(\ell-1)}) \tag{38}$$

and

$$H_{(\ell)} = \Sigma_t^{-1} + y_t^2 \mathbf{x}_t \mathbf{x}_t^\mathcal{T} \cdot p_{t+}^{(\ell-1)} \cdot (1 - p_{t+}^{(\ell-1)}). \tag{39}$$

Then, $\mathbf{u}_{t+1}$ is updated by

$$\mathbf{u}_{t+1}^{(\ell)} = \mathbf{u}_{t+1}^{(\ell-1)} - H_{(\ell)}^{-1} g_\ell. \tag{40}$$

Termination is either when the update on all components of $\mathbf{u}_{t+1}$ is less than some threshold, or after a set number of iterations. Inverting the Hessian $H$ also gives the updated covariance $\Sigma_{t+1}$, whose diagonal elements can be now used to update $\sigma_{i,t+1}^2$, if we apply the algorithm for a sparse problem, where it is infeasible to store all covariances.

Instead of updating $H_\ell$, we can keep track of its inverse $H_{(\ell)}^{-1}$, and there is no need to invert the covariance matrix $\Sigma_t$. With the diagonal form of $\Sigma_t$, all operations can be implemented with linear complexity in $d_t$ using the *Sherman & Morrison (1950)* formula, which simplifies matrix inversions for special matrices. For our specific need here, if $A$ is some matrix, $\alpha$ some constant, and $\mathbf{x}$ some vector, then, the Sherman-Morrison formula is

$$\left(A + \alpha \mathbf{x}\mathbf{x}^\mathcal{T}\right)^{-1} = A^{-1} - \frac{A^{-1}\alpha \mathbf{x}\mathbf{x}^\mathcal{T} A^{-1}}{1 + \alpha \mathbf{x}^\mathcal{T} A^{-1}\mathbf{x}}. \tag{41}$$

Substituting $A = \Sigma_t^{-1}$, $\mathbf{x} = \mathbf{x}_t$, and $\alpha = y_t^2 p_{t+}^{(\ell-1)}(1 - p_{t+}^{(\ell-1)})$, we update $H_{(\ell)}^{-1}$, inverting (39).

As in the marginalization method described in Section 3, we can avoid the iterative Newton method with a first order Taylor approximation of $1 - p_{t+}$ around $1 - \tilde{p}_t$, where $\tilde{p}_t$ is defined in an analogy to (16) as

$$\tilde{p}_t \stackrel{\triangle}{=} \text{Sigma}\left(y_t \mathbf{x}_t^\mathcal{T} \mathbf{u}_t\right) \tag{42}$$

as the un-shrunk prediction of $y_t$ at round $t$ (which is different from $p_t$, which is shrunk by the variance). The approximation leads to the following set of equations to update both $\mathbf{u}_{t+1}$ and $\Sigma_{t+1}$. For simplification, define

$$\mathbf{v}_t = \Sigma_t \mathbf{x}_t. \tag{43}$$

Then, temporarily update $\Sigma_t$, using Sherman-Morrison formula, to

$$\tilde{\Sigma}_{t+1} = \Sigma_t - \frac{y_t^2 \tilde{p}_t(1 - \tilde{p}_t)\Sigma_t \mathbf{x}_t \mathbf{x}_t^\mathcal{T} \Sigma_t^\mathcal{T}}{1 + y_t^2 \tilde{p}_t(1 - \tilde{p}_t)\mathbf{x}_t^\mathcal{T} \Sigma_t \mathbf{x}_t} = \Sigma_t - \frac{y_t^2 \tilde{p}_t(1 - \tilde{p}_t)\mathbf{v}_t \mathbf{v}_t^\mathcal{T}}{1 + y_t^2 \tilde{p}_t(1 - \tilde{p}_t)\mathbf{x}_t^\mathcal{T} \mathbf{v}_t}. \tag{44}$$

Since $\Sigma_t$ is diagonal, the transpose on the last term of the numerator in the first equality is unnecessary. The second equality gives vector multiplications, showing that the complexity is linear in the dimension of the vectors $d_t$. (This is true also to the computation of $\mathbf{v}_t$ when $\Sigma_t$ is diagonal.) Next, $\mathbf{u}_{t+1}$ can be updated

$$\mathbf{u}_{t+1} = \mathbf{u}_t + y_t(1 - \tilde{p}_t)\tilde{\Sigma}_{t+1}\mathbf{x}_t. \tag{45}$$

Now, we can update $p_{t+}$ in (36), using $\mathbf{u}_{t+1}$, and use it to update $\Sigma_{t+1}$ using Sherman-Morrison,

$$\Sigma_{t+1} = \Sigma_t - \frac{y_t^2 p_{t+}(1 - p_{t+})\mathbf{v}_t\mathbf{v}_t^{\mathcal{T}}}{1 + y_t^2 p_{t+}(1 - p_{t+})\mathbf{x}_t^{\mathcal{T}}\mathbf{v}_t}. \tag{46}$$

In the sparse case, we can now take the terms of the diagonal of $\Sigma_{t+1}$ to update $\sigma_{i,t+1}^2$ of the nonzero covariates at round $t$.

Finally, it may be simpler to update the precision matrix $H_{t+1} = \Sigma_{t+1}^{-1}$ instead of the covariance $\Sigma_{t+1}$. Specifically, if multiple updates are performed in a mini batch, the update applied to the covariance cannot be applied additively. However, additive updates on the precision are valid. Thus the updates in (44) and (46) can be replaced by

$$\tilde{H}_{t+1} = H_t + y_t^2\tilde{p}_t(1 - \tilde{p}_t)\mathbf{x}_t\mathbf{x}_t^{\mathcal{T}} \tag{47}$$

and

$$H_{t+1} = H_t + y_t^2 p_{t+}(1 - p_{t+})\mathbf{x}_t\mathbf{x}_t^{\mathcal{T}} \tag{48}$$

respectively. To update $\mathbf{u}_{t+1}$, we still need to invert $\tilde{H}_{t+1}$. We can use (44) if an update was applied to a single round only. If a mini-batch update additively applied multiple updates at once in (47), the updated $\tilde{H}_{t+1}$ must be inverted to obtain $\tilde{\Sigma}_{t+1}$.

The multi-dimensional approach described in this section can be applied to sparse problems, but also to dense problems. In the dense case, the operation in (43) is no longer linear in $d_t$, as the covariance matrix is not necessarily diagonal. The use of Sherman-Morrison formula, however, to invert the precision and covariance, still applies and lowers the complexity of the approach. In the sparse problem, however, this approach may try to force correlations that are not there, that are then ignored. As empirical results suggest, it may not be as good as the marginalization approach because of that. Furthermore, unlike the marginalization approach in Section 3, which achieves best performance if the true prior matches the one used to initialize the algorithm, empirical results demonstrate that the best performances are obtained with priors that are different from the true one with the multi-dimensional method when applied on sparse problems.

## D.2 EXPECTATION PROPAGATION - ASSUMED DENSITY FILTERING

Instead of minimizing the divergence between the approximate $Q$ and the true posterior, we can use the expectation propagation approach, as proposed in Minka (2001), which essentially minimizes the opposite KL divergence, and attempts to match the first two moments. More details can be found in Minka (2001).

## D.3 MARGINALIZED VARIATIONAL BAYES

Instead of using Laplace approximation or matching the location of the peak of the true posterior and the estimated one together with either its curvature or its value, we can apply full VB, by matching the approximate posterior $Q$ with the true one through minimizing the KL divergence $KL(Q||P)$ between $Q$, the approximate posterior, and the true posterior. This requires either iterative approaches, such as mean field approximation EM, or Monte Carlo sampling in order to approximate expectation over a yet unknown $Q$. One can apply this approach in $d_t$ dimensions as the Laplace approximation in Subsection D.1. However, due to the inability to separate the covariates (in the Sigmoid), we would require a power set of samples. If we use $N$ samples per dimension, this approach would use $N^{d_t}$ samples. This can be infeasible and complex if there are a large number $d_t$ of nonzero covariates. Instead, we can use VB only in the $i$th dimension for each feature separately together with the marginalization proposed in Section 3. This can be done by matching the approximate posterior $Q_{i,t}(w_i)$ with the posterior $p(w_i|\mathbf{x}^t, y^t)$ we obtained on the r.h.s. of (12) on feature $i$ after we marginalized on all other features.

The KL divergence can be decomposed into three terms; the KL divergence between $Q$ and the prior $\rho_{i,t}$, the contribution of conditioning the posterior on the probability $p_t$ predicted for $y_t$ , and the log loss (negative log likelihood) term, emerging from the Sigmoid.

$$
KL(Q_{i,t}(W_i)||p(W_i|\mathbf{x}^t,y^t))
$$

$$
= \quad KL(Q_{i,t}||\rho_{i,t}) + E_{Q_{i,t}}\log p_t + E_{Q_{i,t}}\left[\log\left\{1 + \exp\left(-\frac{y_t\left(\mu_{-i,t} + x_{i,t}W_i\right)}{\sqrt{1 + \frac{\pi}{8}\sigma^2_{-i,t}}}\right)\right\}\right]
$$

$$
= \quad E_{Q_{i,t}}\left[\log\frac{\sigma_{i,t}p_t}{\sigma_{i,t+1}} - \frac{(W_i - \mu_{i,t+1})^2}{2\sigma^2_{i,t+1}} + \frac{(W_i - \mu_{i,t})^2}{2\sigma^2_{i,t}} + \right.
$$

$$
\left. \log\left\{1 + \exp\left(-\frac{y_t\left(\mu_{-i,t} + x_{i,t}W_i\right)}{\sqrt{1 + \frac{\pi}{8}\sigma^2_{-i,t}}}\right)\right\}\right]
$$

$$
= \quad \log\frac{\sigma_{i,t}p_t}{\sigma_{i,t+1}} - \frac{1}{2} + \frac{1}{2\sigma^2_{i,t}}\left[\sigma^2_{i,t+1} + \mu^2_{i,t+1} + \mu^2_{i,t} - 2\mu_{i,t}\mu_{i,t+1}\right] +
$$

$$
E_{Q_{i,t}}\left[\log\left\{1 + \exp\left(-\frac{y_t\left(\mu_{-i,t} + x_{i,t}W_i\right)}{\sqrt{1 + \frac{\pi}{8}\sigma^2_{-i,t}}}\right)\right\}\right]. \tag{49}
$$

All expectations are w.r.t. $Q_{i,t}$. The KL term can be computed in closed form, giving the second and third equalities.

The last term on the r.h.s. of (49) cannot be analytically computed without knowledge of the posterior $Q_{i,t}$ at $t$. Instead, we use Monte Carlo, by drawing $N$ samples $S_j \sim \mathcal{N}(0,1)$, letting $W_{i,j} = \mu_{i,t+1} + S_j\sigma_{i,t+1}$. With known $\mu_{i,t+1}$ and $\sigma_{i,t+1}$, we can now approximate the expectation term in (49) as

$$
\frac{1}{N}\sum_{j=1}^{N}\log\left\{1 + \exp\left(-\frac{y_t\left(\mu_{-i,t} + x_{i,t}(\mu_{i,t+1} + s_j\sigma_{i,t+1})\right)}{\sqrt{1 + \frac{\pi}{8}\sigma^2_{-i,t}}}\right)\right\}
$$

where $s_j$ is the $j$th randomly drawn sample. Unfortunately, $\mu_{i,t+1}$ and $\sigma^2_{i,t+1}$ must be updated in this step, and are not known. This requires, again, an iterative update using Newton's method. Similarly to (14) in Section 3, we need to define a prediction $p_{i,j,t+}$ for which the prior (time $t$) means and variances are used for all covariates except the $i$th one, and the updated $\mu_{i,t+1}$ and $\sigma^2_{i,t+1}$ are used for the $i$th mean and variance, respectively. This time, however, this prediction is defined $N$ times, uniquely for each of the $j$th samples

$$
p_{i,j,t+} = \text{Sigma}\left(\frac{y_t\left(\mu_{-i,t} + x_{i,t}(\mu_{i,t+1} + s_j\sigma_{i,t+1})\right)}{\sqrt{1 + \frac{\pi}{8}\sigma^2_{-i,t}}}\right). \tag{50}
$$

Then, the updated mean satisfies

$$
\mu_{i,t+1} = \mu_{i,t} + \frac{y_t x_{i,t}\sigma^2_{i,t}}{\sqrt{1 + \frac{\pi}{8}\sigma^2_{-i,t}}}\cdot\frac{1}{N}\sum_{j=1}^{N}(1 - p_{i,j,t+}). \tag{51}
$$

However, in order to compute both $\mu_{i,t+1}$ and $\sigma_{i,t+1}$, we need to apply Newton's method, optimizing $\mu_{i,t+1}$ and $\sigma_{i,t+1}$ together. We start with $\mu^{(0)}_{i,t+1} = \mu_{i,t}$ and $\sigma^{(0)}_{i,t+1} = \sigma_{i,t}$. For simplicity, we omit the iteration number $(\ell)$ from the notation. The following should be read as updates at iteration $\ell$ which use $p^{(\ell-1)}_{i,j,t+}$ for the updates. For simplicity, let $\alpha_t \triangleq y_t x_{i,t}/\sqrt{1 + (\pi/8)\sigma^2_{-i,t}}$. Then, the joint gradient w.r.t. $\mu_{i,t}$ and $\sigma_{i,t}$ is given by

$$
\mathbf{g} = \left[\begin{array}{c} \frac{\mu_{i,t+1} - \mu_{i,t}}{\sigma^2_{i,t}} - \frac{\alpha_t}{N}\sum_{j=1}^{N}(1 - p_{i,j,t+}) \\ -\frac{1}{\sigma_{i,t+1}} + \frac{\sigma_{i,t+1}}{\sigma^2_{i,t}} - \frac{\alpha_t}{N}\sum_{j=1}^{N}s_j(1 - p_{i,j,t+}) \end{array}\right]. \tag{52}
$$

The joint Hessian is given by

$$
H = \left[\begin{array}{cc}
\frac{1}{\sigma_{i,t}^2} + \frac{\alpha_t^2}{N}\sum_{j=1}^N p_{i,j,t+}(1-p_{i,j,t+}) & \frac{\alpha_t^2}{N}\sum_{j=1}^N s_j p_{i,j,t+}(1-p_{i,j,t+}) \\
\frac{\alpha_t^2}{N}\sum_{j=1}^N s_j p_{i,j,t+}(1-p_{i,j,t+}) & \frac{1}{\sigma_{i,t+1}^2} + \frac{1}{\sigma_{i,t}^2} + \frac{\alpha_t^2}{N}\sum_{j=1}^N s_j^2 p_{i,j,t+}(1-p_{i,j,t+})
\end{array}\right].
$$

(53)

Finally, at iteration $\ell$,

$$
\left[\begin{array}{c} \mu_{i,t+1}^{(\ell)} \\ \sigma_{i,t+1}^{(\ell)} \end{array}\right] = \left[\begin{array}{c} \mu_{i,t+1}^{(\ell-1)} \\ \sigma_{i,t+1}^{(\ell-1)} \end{array}\right] - H^{-1}\mathbf{g}.
$$

(54)

As before, the update terminates if the differences between the values of two iterations are less than some threshold, or after a set number of iterations.

As observed, this method requires $O(d_t N J)$ operations for a single update (and $O(d_t N J T)$ operations overall), where $J$ is the set number of Newton iterations. Empirical results demonstrate that even with $N$ as large as 1000, results are not as good as those with the marginalization with Laplace approximation, presented in Section 3. We note that one can use a two dimensional first order Taylor approximation on $p_{i,j,t+}$ around $p_{i,j,t}$, which is defined similarly to $p_{i,j,t+}$, with the exception of using $\mu_{i,t}$ and $\sigma_{i,t}$ instead of $\mu_{i,t+1}$ and $\sigma_{i,t+1}$, to obtain an approximation for updating $\mu_{i,t+1}$ and $\sigma_{i,t+1}$, as in (17). The approximation should be made for every $j$. It will, however, not require the $O(J)$ operations of the Newton method. The complexity is still of $O(N)$ factor greater than that of the method in Section 3.

# E  ADDITIONAL EMPIRICAL RESULTS

In this appendix, we bring a collection of simulation results demonstrating the performance of Algorithm 1 and the other methods in different settings. In all the simulations we performed we observed that Algorithm 1, with a proper prior for the setting, consistently gives regret close to the lower bound (e.g., $0.5 \log T$ cost for each unknown parameter). The other methods, while in some cases exhibit performance close to that of Algorithm 1, fail to do so consistently in all conditions. The marginalized VB approach requires very large complexity as shown in Table 1 and Fig. 1 to approach the performance of Algorithm 1.

Fig. 4 gives a two dimensional grid of varying $d_t$ and varying true feature weights. In all these simulations, Algorithm 1 with prior matched to the true one, gives minimal regret curves. While SGD seems, with the right choices of parameters, to approach logarithmic regret, its regret is larger, and increases with the true parameters' variance and number of active features $d_t$. Multi-dimensional Gaussian approximation appears to be mis-calibrated on the prior, and depending on the feature density $d_t$ tends to become better only with much larger priors. EP ADF gives larger regrets and while appearing reasonable with higher feature density, seems to be inferior with lower feature densities. Similar results are obtained when replacing the data generation model with models with uniform priors on the feature weights with various ranges.

Fig. 5 shows curves for simulations with different models. On the left, features are nonbinary, and on the right, the model consists of order of magnitude more features, where in average an order of magnitude more features occur in each example. In both cases, Algorithm 1 persists with similar regret rates, whereas other algorithms exhibit larger regrets. With a large number of features, both the multi dimensional Gaussian approximation and SGD have larger regrets, although with even lower prior / learning rates slightly better regret may be possible.

Fig. 6 demonstrates curves of the various algorithms for categorical features, where in each example a fixed set of features are selected from each category. On the right, the selection gives exponentially decaying distribution over the features, so that some features are selected very often while others rarely occur. Again, regret rates behave similarly for Algorithm 1 and SGD. The multi-dimensional Gaussian approximation completely breaks in this setting. This is because it hypothesizes correlations in its updates with features that rarely reoccur.

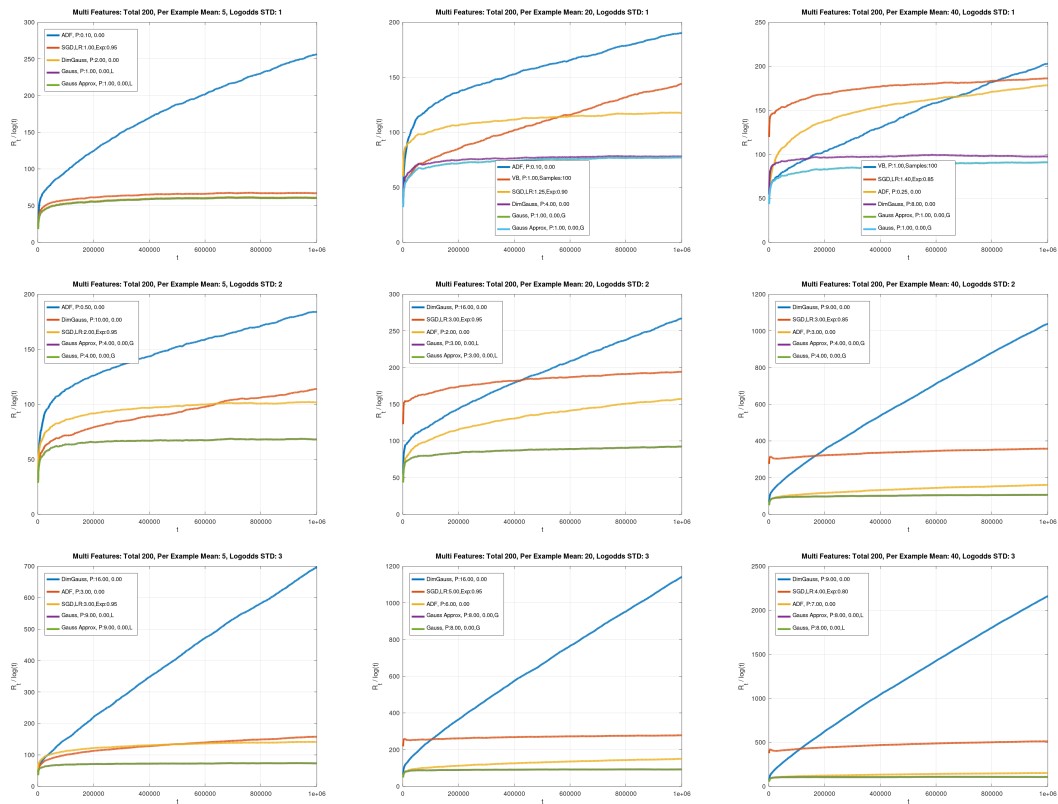

Figure 4: Normalized $\mathcal{R}_t / \log t$ vs. round $t$ for various methods with randomly drawn binary features, with $d$, expected $d_t/d$, and standard deviation of true log-odds noted in each graph. Graphs shown for $d = 200$, $E[d_t] \in \{5, 20, 40\}$ and true log-odds std in $\{1, 2, 3\}$.

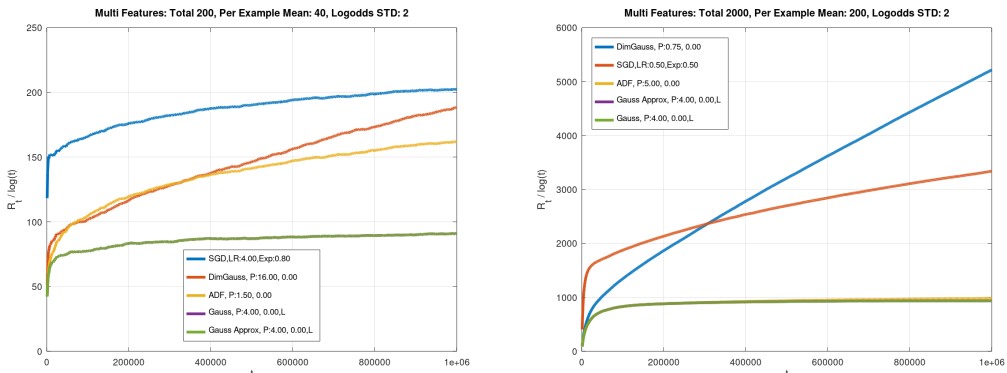

Figure 5: Normalized $\mathcal{R}_t / \log t$ vs. round $t$ for different algorithms and different data generation models. On the left, a model with $d = 200$ features, out of which in average $d_t = 40$ occur in an example, and weight standard deviation is 2, with nonbinary feature values uniform in $[0, 1]$. Right: models with $d = 2000$, average $d_t = 200$, data generation standard deviation of 2, with binary features.

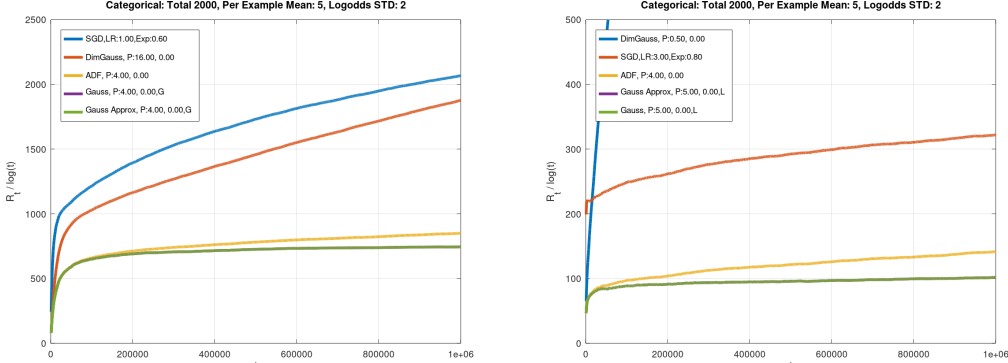

Figure 6: Normalized by $\mathcal{R}_t / \log t$ vs. round $t$ for categorical models and different algorithms. Left: 2000 features in 10 categories of 200 features each, where in each example 5 features from each category are present, with true weight standard deviation of 2. On the right: A similar setting, except that in each category features are selected with a long tail exponential distribution (prioritizing a few features and rarely selecting others).

