# OpenReview forum: "Low Complexity Approximate Bayesian Logistic Regression for Sparse Online Learning"
_ICLR.cc/2021/Conference — Reject_

### Official Review · AnonReviewer2 · 2020-10-25
**Analytical approximate parameter update rules for model paramereters in online Bayesian logistic regression.**

**Rating:** 6
**Confidence:** 3

**Review:**

##########################################################################
Summary:

The paper proposes an algorithm for learning the parameters of a logistic regression model in an online setting. The proposed algorithm is based on two approximations: the posterior at iteration t over the model parameters is assumed to be multivariate Gaussian distribution with a diagonal covariance matrix and the logistic/sigmoid function is approximated by the CDF of the normal distribution. The numerical results show the usefulness of the proposed algorithm.


##########################################################################
Reason for score:

My overall impression is that the proposed algorithm is of sufficient interest to warrant the acceptance of the paper.


##########################################################################
Pros:

1. The paper proposes analytical expressions for updating the parameters of the logistic regression model in an online setting. The computation cost for updating the parameter is reduced.

2. The paper is overall well written and the authors describe in detail each step in the development of the proposed algorithm.

##########################################################################
Cons:

1. For me it is not clear why you settled on a logistic regression problem instead of probit regression and present the logistic regression as an extension. From my understanding, the keystones for the proposed algorithm are the approximation of the prior as a multivariate Gaussian with diagonal covariance and the marginalization step (which itself is due to the Gaussian assumption). From my point of view, the introduction of the extra approximation layer due to the use of the logistic regression is detrimental to the easy understanding of the paper. I believe it would have been better to just introduce a section that describes the modifications to make in order to apply the proposed algorithm to the logistic regression problem.

2. In section 1 on page 2 you make a statement about the role of an approximation in minimizing the the uncertainty. For me it is not clear how the proposed algorithm addresses this statement. Could you please elaborate on the subject? Also, on page 3 you say that it is sufficient to match the component of the posterior that dominates at the horizon. What do you exactly mean by the component of the posterior?


##########################################################################
Miscellaneous:

1. It's welcoming that the proposed algorithm has very good results, however I'm a bit surprised that the results are that good. My wariness comes from the fact that in equation 13 you match an approximation to the posterior distribution, which itself was obtained using an approximation. I would expect the errors due to the approximations to accumulate as more and more data items are processed. However, the regret seems to remain steady as you increase the number of items seen. Do you have an explanation as to why the errors due to the approximations do not accumulate?

2. There are some typos here and there in the article, however I believe you can easily detect them with the help of a spell checker.

---

> ### Author Response · Authors · 2020-11-25
> **Response to AnonReviewer2**
>
> We thank the reviewer for their time, efforts and invaluable feedback.  We revised the paper to address comments by all reviewers.  We address the specific comments made by this reviewer below.
>
> - **Probit vs. logistic regression:** The comment by the reviewer is very relevant, and we agree that the logistic setup introduces more complexity than the probit setup.  The reason we chose to focus on logistic regression is mainly the larger popularity of logistic regression in practice.  One could argue about the justification of this popularity, but this is the case, especially in the machine learning literature and applied community.  We also note that with probit regression, numerical tables or approximations are necessary to compute the Gaussian CDF.  A decision could go either way here - the advantage of logistic regression is its popularity, and that of probit is its simplicity.  We chose for the first, but we also extended the discussion on the second in a new appendix in the revised version.
>
> - **Approximation and Uncertainty:** One of the points we tried to emphasize in the paper is the seminal relation established by Rissanen between the minimum description length (MDL) and prediction error. Regret is a component of the MDL, which corresponds to the epistemic uncertainty, or in other words, to how well the parameters of the true model are estimated.  Therefore, an algorithm with lower regret also estimates the unknown model parameters better (with lower prediction error).  It has been established (Kakade, Ng. 2005, Foster 2018, Shamir 2020) that Bayesian methods in theory can achieve the lower bounds on regret.  Therefore, the role of a good approximation is to retain that property, and degrade only components of the distribution that do not affect the regret (or that minimally affect the regret).  The claim was made in the paper to pave the path to demonstrate that there are two ends to this.  One, use an approximation that is good enough, but on the other hand, don’t use an approximation that is much more complex, but adds no value over a simple approximation in what matters.
>
> - **Match components of posterior:** The message in this claim is that an approximation should not scale down the effect of the prior at the point of the distribution where the algorithm converges to at the end.  As epistemic uncertainty becomes smaller due to more data, the distribution of a model parameter becomes more and more like a point mass around the regret minimizing value of the parameter.  A bad approximation reduces the magnitude at this point.  As shown in (Kakade, Ng. 2005, Shamir 2020), the optimal regret achieves for a parameter is roughly the logarithm of the prior at this point (or an integral around this point).  If the approximate posterior scales down the magnitude at this point or its vicinity from the true one, there would be additional regret losses.  Given all that, the idea is that we try to pick approximations that have a property that they do not scale down the posteriors at points that are likely to hand up being the convergence point of the parameter.  Simple approximations,  like Laplace, that focus on the peak follow this. However, complex approximations, like VB, give too much weight to points far from the peak by trying to match the full posterior, and by that also harm the regret.
>
> - **Accumulation of approximation errors:**  We believe that this happens, and that there are some additional penalty terms to the regret.  However, if you look at the approximations more carefully, while they lose in some regions, they gain in others.  Attributing to the previous point, the approximation can scale down the effect of the prior in one example, but then correct it and scale it up in a subsequent example. The overall effect may hide some of these losses by correction gains that happen in the subsequent examples.  It is very difficult to derive regret bounds for these approximation approaches, but such bounds could explain this better.
>
> - **Typos:** We are making several passes to fix typos on the revised submission.

---

### Official Review · AnonReviewer1 · 2020-10-27
**Interesting paper but the experimentation is too limited. Several contribution claims appear debatable and requires further clarification.**

**Rating:** 4
**Confidence:** 4

**Review:**

PAPER SUMMARY

This paper proposes a new posterior approximation scheme for probabilistic logistic regression. In the online learning setup, the proposed approximation leads to a closed-form parameter update. In particular, the key ideas here include:

(1) A probit approximation of the sigmoid likelihood in Eq. (5). This, along with the assumed statistical independence among individual features and another integral approximation trick used in (Murphy, 2012), leads to a posterior representation in forms of a normalized product between a sigmoid and a normal PDF.

(2) Matching the location and density of its mode to that of another Gaussian surrogate to yield the closed-form update that expresses the parameter at round (t + 1) in terms of its counterpart at round (t) analytically in Eq. (15)

NOVELTY & SIGNIFICANCE

If I understand correctly, the main arguments in favor of the proposed method are: (a) the iterative update derived from Eq. (15) only involves a single sample at round (t) which is more efficient than VB's or EM's stochastic gradient updates that require re-iterating over multiple mini-batches of past data; and (b) the new approximation scheme performs empirically better than existing method, and matches state-of-the-art regret lower-bound.

Given the current form of the manuscript, these arguments, however, are not very convincing to me as I elaborated in the followings:

First, regarding (a), while it is true that VB's or EM's offline updates are more expensive than the update derived from Eq. (15), it seems to me they can still be made more efficient in online setting with very minor modifications. For instance, in both VB and EM, the lower-bound function often factorizes additively across data point so if we cache all gradient computation in step (t), the computation of new gradient in step (t + 1) also involves only the latest data sample which is likely not more expensive than iteratively solving Eq. (15) -- I would like to hear the authors' thoughts on this.

Second, the claim that the new approximation scheme performs better than existing methods is not well-supported. In this regard, I wonder why the improved loss is only demonstrated in a very controlled, synthetic experiment settings. One could always argue that as far as only synthetic experiments are involved, there might exist other settings for which we might see the opposite. To me, positive results on a single set of synthetic experiments often do not speak conclusively to whether one method would be more useful than another in practice -- a better way to demonstrate this is to also evaluate the proposed method on real-world benchmark datasets (especially those extracted from practical domains that were mentioned in the introduction)

Furthermore, the evaluation does not seem to include all the most relevant baselines (more on this later) as well as processing time comparison. Given that one of the key contribution claims here is the computation advantage over EM's and VB's, both theoretical complexity and empirical demonstration of averaged running time should be provided.

In addition, I also find part of the positioning of this paper somewhat misleading, especially when it criticizes stochastic gradient methods for incurring additional losses while the target posterior keeps moving with subsequent samples. Isn't this also true for the proposed method? While Eq. (15) is analytic, solving it for the parameter update is clearly not analytically tractable which also requires iterative update; and then even the resulting approximated posterior has to be matched again to another Gaussian surrogate. Lastly, the conclusion surprisingly summarizes that with proper prior, the method "matches regret lower-bound" -- this seems like an overclaim as there is no theoretical analysis in the paper to back this up.

TECHNICAL SOUNDNESS

I have gone through the derivations and I do not spot any serious issues, but there are parts that seem to lack justification. For instance, could the authors please detail the steps that lead to Eq. (5)? Also, it will be even better if the authors could provide plots to visually demonstrate how the derivative of sigmoid is close to the scaled normal CDF.

I also find it a bit strange that this paper highlights the sparse aspect of data as an issue but in the end, it appears more to be a necessary setting that justifies the assumed diagonal form of the variance matrix, which is a necessity for most of the technical derivations in this paper.

CLARITY

While the paper is sufficiently clear, I find the introduction is a bit too long. It could have made its points using much less space. In the current form, it kind of distracts the readers from those key points by throwing in many subsidiary positionings that are better discussed & organized in related work.

The claim on analytic update is also a bit exaggerating as solving Eq. (15) for an update equation is after all not tractable and we will have to resort to iterative method such as Newton's or Taylor series expansion anyway.

EXPERIMENT

On the experimental evaluation, my key concern (as mentioned above) is the lack of evaluation of real-world dataset, and in addition, some of the more recent methods such as (Nguyen, 2017a; 2017b) were not included in the baseline.

The presentation of the evaluation is also problematic when the results seem to be collected on only 1 single run. As a standard practice, please re-run the same experiment multiple times and report the margin of error. It is also unclear why the Y-axis is t/log(t) -- first of all, do you mean r_t /log(t) instead?; and secondly, why do we divide it by log(t) but not t? Aren't we ultimately interested in the limit of the average regret r_t/t?

Furthermore, despite the claim that this method is proposed for the setting with huge feature sets (with billions of dimensions), the synthetic experiment is mostly around 200; and up to 2000, which is a bit disappointing. On the same note, despite having a main claim of computational efficiency, there is not a single experiment that showcase this advantage over existing methods.

REVIEW SUMMARY

The paper aims to develop a posterior approximation scheme for sparse online logistic regression. While the key idea is to achieve faster computation time and better regret rate than existing methods, this point has not been demonstrated sufficiently due to a complete lack of complexity analysis and experiments on large-scale, real-world data with more recent benchmark. The paper also appears to overclaim on the analytic tractability of its update as I have pointed out above.

Note: This is only my preliminary assessment and there is of course the possibility that I might miss some important points. In that case, I am looking forward to receiving detailed clarifications from the authors.

---

> ### Author Response · Authors · 2020-11-25
> **Response to AnonReviewer1 - Part 1**
>
> We thank the reviewer for their time and effort, and for their invaluable comments.  We revised the paper to incorporate changes motivated by the reviewer’s suggestions.  Below are specific responses to some comments.
>
> We believe that some of the reviewer’s comments originate from some misconception about the content of the paper.  Below we try to clarify these misconceptions and the differences between the proposed methods and some claims made by the reviewer.
>
> - Approximating of posterior in this paper is a tool for minimizing regret with a Bayesian method, and not the other way around.  The purpose is not to approximate the posterior, but to minimize regret.  In that, this work is different from much of the work in the literature that focused on matching the posterior.  It also differs from most work by doing this online, where the problem dictates having a prediction based only on data that was observed so far.  Most literature on Bayesian methods considered the batch setting, where a batch of (training) data is available as long as needed to update and refine predictions.  This is not the case here.  A datum is seen, the model must update, and move on to the next datum.
>
> - **Key ideas:** The reviewer identifies technical ideas that were used for the analysis.  However, these are not the conceptual key ideas that motivated this work, these are technical steps that were used for the analysis. The key ideas are: A) a properly designed Bayesian mixture over the parameter space retains the effects of the optimal (regret minimizing) values of the parameters, and by doing so, yields optimal regret.  B) Retaining the effect of such a mixture at the optimal point by an appropriate low-complexity approximation can retain regret optimality.  Statistical independence allows retaining this effect (or most of it) with a diagonal approximation.  Such statistical independence is justified in sparse problems unlike dense problems, where more features are correlated, or co-occur in the same examples, in the sparse problems co-occurrence is more sporadic, and then diagonalization is justified.
>
> - **Novelty:** “the iterative update derived from Eq. (15) only involves a single sample at round (t)” - This is NOT the message in the paper, this statement distinguishes an online scheme from a batch method.  The problem considered in this paper is indeed the online problem, and most of the literature considered the batch one.  However, the main aspect distinguishing this work from VB methods (even online ones) is the fact that the update per-example in the online setting requires O(1) operations, whereas VB methods require O(N) where N is the number of Monte Carlo samples drawn to apply an update for the current example.  The (b) claim made by the reviewer is true.  Despite the low-complexity the method outperforms other low-complexity or high complexity methods for the sparse problem.
>
> - **Novelty comment (a) reviewer elaboration:** The reviewer considers updates on multiple examples vs. update on a single example.  In our setting, updates for all algorithms are applied for one example.  We do show a VB based approach that applies such updates for comparison.  In the runtime benchmarks that we added, it is shown that the algorithms with VB take orders of magnitude more time than our approach, only to give inferior results.
>
> - VB, EM, CAVI - all these algorithms are more expensive than our proposed method, as they either iterate over the features (not necessarily over examples) or require multiple samples to average over.  They can be applied in the online setting, as we illustrate for VB in the paper, but their runtime/performance tradeoffs are inferior.  And at best, since they are based on the same theory foundation of our method, they can achieve the same regret as our method with much higher complexity.
>
> - While loss can factorize across different data points, it does not across the features of the same data point, because the label prediction is a function of the weighted sum of covariate weights. This means that to apply an approach like VB over an example with 30 active features, one would need to average over X^30 samples, where X is the number of samples to get a good estimate of one dimension of the posterior.  So, while we can factorize over examples, this is not the issue here.  Factorizing over covariates of the same example is the issue.  However, even with marginalization, as we show in the paper, applying 1000 samples to each dimension after marginalization still falls short of our method in terms of regret, but results in an algorithm whose runtime is O(1000) slower.

---

> > ### Author Response · Authors · 2020-11-25
> > **Response to AnonReviewer1 - Part 2**
> >
> > - **Synthetic vs. Real data:** The paper addresses a problem that, to the best of our knowledge, has not been studied in prior literature, i.e., regret for **practical** Bayesian algorithms for online learning. We are unfamiliar with any results measuring regret for such methods.  While benchmark datasets are useful in measuring loss of one algorithm vs. another, they are not useful in measuring regret, as we have no knowledge of ground truth.  Furthermore, machine learning models on real datasets incur misspecification, and real datasets may be nonstationary.  Our goal in this paper is to show that the method behaves favorably in terms of regret, and isolate out all the other effects.  For that, theoretical guarantees and synthetic data are the right method.  We are unfamiliar with any theoretical regret guarantees on any practical Bayesian approximation methods, and we believe we are the first to provide regret results for such algorithms.  (Note that there are papers (Kakade, Ng. 2005, Foster 2018, Shamir 2020) that provide such bounds for theoretical methods, but not for practical ones).
> >
> > - “One could always argue that as far as only synthetic experiments are involved, there might exist other settings for which we might see the opposite” - This claim is even more true for real datasets.  The experiments we reported on synthetic datasets tried to give extensive coverage to many different statistical models of the data, an attribute that is impossible with real benchmark datasets.
> >
> > - **Public datasets:** We have added results on a public dataset to illustrate that despite the issues mentioned above, we still attain favorable results. As the reviewer suggests, the dataset is one from a domain we consider.
> >
> > - **Runtime:** We included runtime measurements of the algorithms, that are not surprising and match the complexity of the described algorithms.  They clearly show the advantage over the VB methods.
> >
> > - The paper “criticizes stochastic gradient methods for incurring additional losses while the target posterior keeps moving with subsequent samples.”: The paper **does not criticize SGD** while the posterior keeps moving, it criticizes variational techniques that take SGD steps instead of fully computing the posterior at every update.  In such cases, the target keeps moving before the algorithm reached the previous target.  This is not true for eq. (15), as all iterations are performed before the next example is observed.
> >
> > - **Eq. (15), analytical update, and eq. (17):** “The claim on analytic update is also a bit exaggerating as solving Eq. (15) for an update equation is after all not tractable and we will have to resort to iterative method such as Newton's or Taylor series expansion anyway.”: Eq. (15) does not have a closed form solution.  However, A) we note the relation the Lambert function, which leads to a numerical solution, B) Applying Newton’s method is only for the mere purpose of refining a solution for an equation that does not have a closed form solution.  This is very different from the optimization iterations that need to find a common solution to many points, such as in the CAVI algorithm.  In fact, convergence occurs rather instantly, and the runtime using the update with eq. (15) is practically identical to the one with (17). C) The version of the algorithm with (17) applies a single update per feature per example, requires no iterations, and gives the same performance as that of (15).  This is clearly superior to the VB methods.  Again, it should be noted that all of the updates (including VB) are performed per example, and there is no iteration over different examples.  The resulting posterior does not “have to be matched again to another Gaussian surrogate”, eq. (15) **is the matching**.
> >
> > - “The paper also appears to overclaim on the analytic tractability of its update” - as we explained above, this claim by the reviewer is incorrect.
> >
> > - We agree that the conclusion should be toned down to only claim that empirical results match the lower bounds.  We revised the paper accordingly.
> >
> > - An appendix was added to demonstrate the approximation in Eq. (5).  (We had this in the original text, but omitted it due to the page limit.)

---

> > > ### Author Response · Authors · 2020-11-25
> > > **Response to AnonReviewer1 - Part 3**
> > >
> > > - **Highlighting sparse aspect:** We respectfully disagree with the claim made by the reviewer.  For a dense problem, one would apply dense updates on the full covariance matrix.  However, such approaches, as we show with the multi-dimensional updates for some of the results (including new ones for the Criteo dataset and for long tail distributions with millions of features) become inferior in the sparse settings, as they try to explain correlations that are not there.  As theoretical results (Kakade, Ng. 2005, Shamir 2020) show, yes, if we keep the full dense covariance of all features that exist, update this matrix for each example and retain the form of the posterior, whatever it is, we will achieve the regret lower bounds.  However, such an approach is not feasible in practical problems where there are billions of features. In fact, it is at the limit even with only hundreds of features.  Even with dense problems with such a small number of features, the Gaussian approximation of the posterior already deviates away from the true posterior.  If one considers a practical problem with billions of features out of which only a few tens occur at a given example, the likelihood of uncorrelated features to co-occur is very low, and the sparse approach is a very reasonable approximation.  For correlated features that co-occur multiple times, the multi-dimensional approach in Appendix D.1 could capture the correlation, and we do observe in the empirical results for not-so-sparse cases (200 features, 40 of which show in each example), that this method becomes more accurate.
> > >
> > > - “I find the introduction is a bit too long”: See comment by AnonReviewer3: “The introduction properly positions the work” - We invested a huge amount of effort in trying to position the paper in an ocean of related literature, and we tend to agree here with AnonReviewer3.
> > >
> > > - **Variational Continual Learning:** The work in (Nguyen, 2017a, 2017b) is related, but the algorithms are not relevant to compare to our method as they address a different problem.  The main focus in these papers is for the system to track changes in the underlying distribution and adapt to them if needed, while keeping the knowledge of past examples if changes have not occurred. Our setup is for a stationary problem, where the underlying governing distribution does not change.  Furthermore, the methods in these papers are focused on dense updates, and are not feasible for the sparse problem. We included several algorithms that fit in the same problem setup for comparison purposes, as our empirical results demonstrate.  Note, though, that for none of these approaches could we find regret results in the relevant references.
> > >
> > > - **Single run:** As we evaluate regret, a single run is sufficient to measure it.  One could run multiple different runs, but if performed on a sufficient number of samples, with ergodicity and stationarity, the total regret would not change much.  Furthermore, results on a benchmark dataset are also on a single set of data points.
> > >
> > > - **Regret normalization by log t:** The goal is to compare the regret to the lower bounds which are logarithmic in t.  As we explain in the paper, normalizing by log t gives a constant at the horizon if the true regret is logarithmic. This is the purpose of this normalization.  Note that normalizing by t would be counter productive, as it will let the regret diminish, and will not distinguish between the different algorithms, as long as they all have regret rates which diminish relative to the data uncertainty.
> > >
> > > - Y-axis - yes, this is a typo, it should be R_t / log(t).  We are correcting this in the revision.
> > >
> > > - **Huge feature sets:** We are incorporating results on the Criteo dataset and synthetic data with millions of features.  Such settings favor the sparse approach, as the likelihood of more features co-occurring is lower.  The issue, however, is that if such a setting is simulated, more data points are needed to see the same effects.  In practice, what is relevant is the number of features per example, and its ratio to the total number of features.  The assumption is that for a given example a small number of features is present (which is true in practice for such systems).  With a larger number of total features, and the same number per example, we see the same effect as we see if we lower the total as well as the per example count.

---

### Official Review · AnonReviewer3 · 2020-10-28
**Fast approximate Bayesian algorithm but no guarantees on the quality of the approximations and no convincing experiments**

**Rating:** 4
**Confidence:** 4

**Review:**

This paper proposes an algorithm  for online logistic regression  based on analytical formulas that approximate the Bayesian predictive posterior. These approximations are based on assuming a diagonal covariance Gaussian form for the posterior at each iteration that is optimized to fit the true posterior. Two alternatives are proposed for the optimization: one based on Newton's method and the other based on a Taylor series approximation, which experimentally yields similar results. Thanks to these closed formula approximations, the resulting algorithm has a constant low cost per observation and  is particularly suitable for sparse high-dimensional scenarios. It is empirically shown that the algorithm achieves a known regret lower bound on synthetic data when the true prior on the weights used to generate the data is provided to the algorithm.

Strong points:
- The introduction properly positions the work
- A low-complexity algorithm for online prediction suitable for sparse high-dimensional features is proposed

Weak points:
- No theoretical guarantees on the regret
- Regret is strongly dependent on the prior
- No experiments on real datasets


This paper proposes a low-complexity approximated bayesian algorithm that aims at attaining known regret lower bounds that have been recently shown [Shamir 2020] to be achievable by exact Bayesian methods. Unfortunately, no theoretical guarantees are provided on the impact of such approximations and the empirical performance is reported to be strongly affected by the choice of the prior. In particular, it is reported that the prior has to fit the data for good regret. This strong dependence on the prior makes it of limited practical interest in real life online scenarios. Therefore, I think that the paper is not strong enough and recommend its rejection.

Detailed comments:

Section 3 could be better structured and a "road map" at the beginning of the section would help the reader to quickly understand the different steps involved in the approximation.

If I understood correctly, in order to build a synthetic dataset,
- you sample the weights $w^\star$ using some prior (a gaussian with mean 0 and STD specified in the title of each plot).\
- At time $t$, you sample random features $x_{i,t}$ using different distributions, depending on the type.\
- You add up these features weighted by $w^\star$ to obtain the true log-odds that is used to define the probability distribution (through the sigmoid) from which $y_t$ is sampled.

Then, the following sentence is confusing:
"The “true” log-odds of the features drawn at t were added up, weighted by $x_{i,t}$, and converted to probability with the Sigmoid function."

In the following sentence you use $\tau$ that doesn't appear in eq. 4:
"Results describe a time series on (4), where summation is on $\tau$ up to example $t$ at time $t$."

"We demonstrate that it is sufficient to approximate the component of the posterior which will dominate
at the horizon, matching it by a diagonal Gaussian approximation ,"
How is this demonstrated?

Eq 11: (d) should be with $\approx$ instead of $=$

Since nonparametric Bayesian methods are mentioned in the abstract, it would be interesting to see a comparison against online nonparametric Bayesian methods.

In my opinion, the title should say "online prediction" instead of "online learning", since the latter does not necessarily imply predicting at each round and cumulating the loss.

Typos:\
 -- mostly a function *of* the number\
 -- Jeffery's prior -> Jeffreys prior


=====POST-REBUTTAL COMMENTS========

I thank the authors for the response and the efforts in the updated draft, in particular with respect to the new experiment on the Criteo dataset.
Regarding the following arguments:

1- *"The dependence of regret on the prior applies to any Bayesian method, in the same way that tuning the learning rate for gradient methods is necessary."*

I agree that, for batch methods, the prior can be tuned to the data. Nevertheless, in an online prediction setting, the prior needs to be chosen before observing the data and once you start predicting you can't go back and change the prior for the already predicted sequence.

2- *"To the best of our knowledge, none of the literature that proposed practical (online) Bayesian methods proposed regret bounds or even studied regret for these methods"*

Regarding both arguments, there is recent literature that proposes a practical nonparametric online prediction method with regret guarantees for a fixed given prior:

*Lhéritier, Alix, and Frederic Cazals. "Low-Complexity Nonparametric Bayesian Online Prediction with Universal Guarantees." Advances in Neural Information Processing Systems. 2019.*

Therefore, unfortunately, my first two concerns (theoretical guarantees on the regret and strong dependence on the prior) remain so I retain my original decision.

---

> ### Author Response · Authors · 2020-11-25
> **Response to AnonReviewer3**
>
> We thank the reviewer for their time and effort, and for their invaluable comments.  We revised the paper to incorporate changes motivated by the reviewer’s suggestions.  Below are specific responses to some comments.
>
> - **Theoretical guarantees:** To the best of our knowledge, none of the literature that proposed practical (online) Bayesian methods proposed regret bounds or even studied regret for these methods.  Loss and accuracy were considered, but it was never indicative which fraction of the loss to attribute to data uncertainty, and which to regret (which corresponds to the model/epistemic uncertainty). In fact, we believe that such study in our paper is novel.  While several theoretical papers (Kakade, Ng. 2005, Foster et. al. 2018, Shamir 2020) showed that Bayesian methods can, in fact, be optimal for regret, the literature on practical approaches lacks such connections.  Our paper intends to bring a proof of concept to a low complexity approach that appears to outperform other methods or match substantially more complex methods.  Proving regret bounds for such an approach could be a theoretical COLT paper by itself, and no such bounds exist for any other practical methods.  We believe that synthetic data is the first proof of concept to regret performance of such methods.  The synthetic data brings a dimension that real data does not, by allowing empirical results to focus on regret, regardless of the true parameter values, which modify the loss.  It also isolates the measurements from other phenomena that affect real data, such as model misspecification and nonstationarity.
>
> - **Regret and prior:**  The dependence of regret on the prior applies to any Bayesian method, in the same way that tuning the learning rate for gradient methods is necessary.  While theoretical results (Shamir 20) do show that the lower bounds are achievable regardless of the prior (as long as a comparator is confined to some space), performance of practical techniques is a function of the prior chosen.  What we show is that if the prior used matches the data, we achieve better regret than other methods whose hyper-parameters were also tuned to attain their best performance.  While it would be nice to have an algorithm whose performance is not affected by the choice of either prior or other hyper-parameters, no other known methods have such properties. We respectfully disagree with the reviewer’s claim that “This strong dependence on the prior makes it of limited practical interest in real life online scenarios”.  Practical methods **do tune** learning rates and/or priors to fit the data in practical systems.  Our algorithm is no different from that.  Once tuned, it outperforms other methods.
>
> - **Real datasets:** As discussed above, it is nice to have results on real datasets, and we added such results. However, loss on these datasets reveals limited information about regret, which is what the paper attempts to minimize. Loss, relative to other methods, does indicate whether regret is better for one algorithm over another, but it doesn’t measure the regret.  Furthermore, loss on real data can also be affected by properties of the data, like nonstationarity, which we do not attempt to address in this paper. To address nonstationarity, misspecification or other artifacts of real data, additional modifications may be necessary to an algorithm.  The goal of the paper is to show that this method works in a clean stationary setting. This can lead to development of methods based on the foundation laid by this paper that address other properties of learning models on real data.
>
> - A road map has been added in Section 3.
>
> - Sentences that were unclear to the reviewer were modified, and minor corrections suggested were made.
>
> - “"We demonstrate that it is sufficient to approximate the component of the posterior which will dominate at the horizon, matching it by a diagonal Gaussian approximation ," How is this demonstrated?”  - Our regret results show that the proposed algorithms are as good as or outperform methods that match the full distribution instead of the important points of the distribution.
>
> - "Since nonparametric Bayesian methods are mentioned in the abstract, it would be interesting to see a comparison against online nonparametric Bayesian methods.” - There is no mention of nonparametric methods.  The reviewer refers to a comment about hyper-parameters that need to be tuned.  For the theoretical Bayesian methods (with known comparator class) no such hyper-parameters need to be tuned.
>
> - We respectfully disagree with the name change proposed by the reviewer.  The proposed methods is an online learning algorithm.  We measure regret in the standard way it is measured in the online learning literature, and the algorithms learn a distribution within the uncertainty of the parameters.

---

> ### Author Response · Authors · 2021-02-13
> **Response to Post Rebuttal Comments by AnonReviewer3 - Part 1**
>
> **Batch Vs. Online**
>
> - We agree with the reviewer that once one starts predicting, they can’t go back to change the prior.  However, practical online systems train many models over the same training data, or over a sample of the full training data.  They study the dataset (the existing subset of the dataset) and tune hyperparameters to that dataset.  This is how popular SGD/AdaGrad are used in practical systems.  Once optimal values of the hyperparameters are found, the model is trained with these hyper-parameters, first on data that is already available, and then on newly incoming data.  The optimal value of the learning rate (scaling factor, for example, in standard AdaGrad) **is a function of the true data model parameters**, and if other values are used, there are clear degradations in performance.  We would like to reiterate that this is common practice in realistic practical systems, and thus our results are very useful here.
>
> - Furthermore, even if one sees this as a drawback, the paper shows results that are not known in the literature, and lays the foundation to address the problem raised in subsequent work.
>
> **Irrelevance of the paper noted by reviewer to the problem we study**
>
> *"there is recent literature that proposes a practical nonparametric online prediction method with regret guarantees for a fixed given prior”* - While the paper referenced is very interesting, we respectfully disagree with the relevance of this reference to our work.  The problem they considered is very different from our problem.  It is a dense problem, where feature values are required to be distributed (uniformly) in [0,1].  They partition the d-dimensional feature space into a set of partitioning.  Using the CTW algorithm, for each partitioning, a mixture is performed over all possible subpartitioning along the partition path.  Using the KT probability estimator, a mixture over the governing probability is performed for each individual partition, which allows the online update.  The eventual prediction is on a mixture over all partitionings represented by a partition tree.  Each step of partitioning merely determines if marginalization over the given dimension is sufficient, or different values of the feature affect the prediction sufficiently, such that the feature value should be considered.  The results hold **ONLY IF**
> - The discretization is sufficiently fine, such that the losses due to approximating the parameters as fixed in every region is negligible.
> - The distribution over the parameters leads to a nice distribution over [0,1] of the feature values, and
> - The partitioning covers the space nicely, and does not grow with the horizon.
>
> Below (and in part 2) are more details to these points:
> - Their results are implicitly dependent on the actual depth of the partitioning for both complexity and regret.  If the nice assumptions about the feature value distribution do not hold there IS loss of generality, and |A| can grow fast, and lead to non-diminishing regret, as well is infeasible complexity.
> - None of the assumptions they need are satisfied for our problem.  An example of our problem is where there are millions of features only few of which show in each example, and some show very few times in billions of data points.  In such a case, the partition in the proposed paper is guaranteed to become of the same order of magnitude as the horizon, because in our setting the same combination of features almost never shows up in more than a single example.  Therefore, the depth of the context tree will become O(d), which will result in complexity O(dT) instead of O(d_t T) for the full algorithm, where d_t << d.  Furthemore, the logarithmic term in n in (8) in their referenced paper, will have a coefficient that is omega(d) which will dominate the regret, and the conditions of Corollary 1 will not hold, so that the regret will have additional term from the discretization that is not negligible.
> - More precisely, if we consider a binary feature problem, where a feature either exists with z = 1, according to their notation, or does not exist (z=0), then what their approach does is implicitly learn a parameter for each of the possibly 2^d parameter combinations (where the parameter is learned with the KT estimator, and if some dimension is non-informative, the context tree will eliminate its dimension through mixing with a lower dimension combination).  In cases all parameters are informative, both complexity and regret will behave as 2^d (if d is O(log n)).  If d is larger, it will behave like learning a parameter for each example, as a different combination of parameters is likely to show for each example.  This will not yield diminishing regret at all.  This is because their algorithm does not distinguish between covariates, which is key to logistic regression problems.  In that, the problem studied in that paper is substantially different from the problem we study, and the loss bounds are irrelevant to our work.

---

> > ### Author Response · Authors · 2021-02-13
> > **Response to Post Rebuttal Comments by AnonReviewer3 - Part 2**
> >
> > - The paper referenced by the reviewer does not empirically show regret results, it shows normalized log loss.  And the results clearly demonstrate a non-negligible gap between the entropy and the algorithm loss, which clearly suggest higher regret than logarithmic.
> >
> > - The claims in that paper are probabilistic depending on nicely behaved distributions of the features and labels. Regret that we study is defined per sequence with no probabilistic arguments.  Furthermore, the need to do an ensemble of partitionings in the referenced paper suggests that another Bayesian mixture on top is, in practice, necessary to offset the losses caused by data not behaving as expected for a specific dataset, suboptimal random selection of the partitioning.  In fact, the algorithm presented in the referenced paper is sensitive to the order in which the partitioning is selected.  If the least informative features are selected to be partitioned on first, this will lead to degradation of performance, which in the simulations is “hidden” by the ensembling of CTW ensembles.

---

> > > ### Comment · AnonReviewer3 · 2021-02-16
> > > **Response to Post Rebuttal comments**
> > >
> > > Thank you for your detailed response.
> > >
> > > **"Batch Vs. Online"**:
> > > I understand your practical point of view. For future submissions, I suggest you to emphasize that you have in mind scenarios where you are allowed to train the model on some part of the data, for which no predictions are required.
> > >
> > > **"Irrelevance of the paper noted by reviewer to the problem we study"**:
> > > I agree that the results of that paper are of a probabilistic and asymptotic nature but the only assumption that is made is that the features follow a continuous distribution in Rd---not necessarily uniform [0,1].
> > > (Since the splits are axis-perpendicular, by monotone transformation invariance, the analysis reduces to the case with uniformly [0,1] distributed features.)  Therefore, I agree that the method of that paper is not suitable for binary or sparse features and that the reference is irrelevant for those cases.

---

> > > > ### Author Response · Authors · 2021-02-21
> > > > **We thank AnonReviewer3 for the additional clarifications!**
> > > >
> > > > We thank AnonReveiwer3 for the latest response on our rebuttal.  We greatly appreciate the time and effort the reviewer has spent on this paper, and especially on taking the time to respond to our comments post decision.

---

### Official Review · AnonReviewer4 · 2020-10-29
**Interesting, but more work to be done**

**Rating:** 4
**Confidence:** 4

**Review:**

The authors propose a low complexity approximation method with closed analytic forms for doing logistic regression in the sparse, online setting. They first introduce the marginalized bayesian gaussian approximation approach, which essentially replaces the sigmoidal with a gaussian. They then give approximate expressions for the prediction and marginalization terms, as well as ways to approximate/update the posterior.

This is an interesting approach. The pros are clear: closed-form analytic updates that attempt to retain the core advantages of a Bayesian approach.

The authors should address the following questions/concerns:

Assumptions: The diagonal Gaussian assumption appears to be quite strong. Are there comments/extensions on how to incorporate this for correlated/collinear covariates?

Approximation: The approximation is based on a Gaussian approximation to the sigmoidal. Are there any theoretical/quantitative guarantees on the approximation error? Right now the approximation is just a heuristic.

Experiments: since one of the main proposed advantage of this scheme is that it is computationally faster, it would be nice to have experiments/charts showing how much faster (in actual experiments) does this method run when compared to competitors like VB or a Gibbs sampler. Also, there is a Newton’s method step in the algorithm. Could the authors comment on how that affects run time?

Motivation: If you are using a Gaussian to approximate the Sigmoidal, since you are using a Gaussian anyways, why not just use an online probit regression then, instead of the logistic?

Overall, I think this is a potentially useful proposal that is interesting, but I think a) more experiments needs to be done with competitors to illustrate not just regret, but runtime comparisons b) there needs to be more theoretical rigor as to how an approximation performs and what the approximation error/bounds/guarantees are.

I also looked at the citations more closely, and I discovered that there are some mis-citations:
For example, you cited Terry Anderson. The theory and practice of online learning. Athabasca University Press, 2008, which upon further inspection is about online (as in over-the-internet) learning, not online learning in the machine learning sense.

---

> ### Author Response · Authors · 2020-11-25
> **Response to AnonReviewer4 - part 1**
>
> We thank the reviewer for their time, efforts and invaluable comments.  We revised the paper to incorporate changes motivated by the suggestions of the reviewer.
>
> - **Diagonal assumption:** This assumption is standard in **sparse** problems, and in the Bayesian literature of such problems (as variational inference methods), where full matrix operations are not feasible due to the dimensionality of the problem. It is also standard for similar convex optimization techniques using gradient methods. The problem we consider can allow billions of features, out of which only a different small subset is present in each example. In practice, the sparse diagonal solution works well for large scale datasets.  (Diagonalization is, in fact, the approach used in many practical systems that use gradient methods.) We add results on the Criteo benchmark dataset to demonstrate this. Furthermore, approaches like the one we empirically compare to, that update the full matrix of nonzero covariates at a given example, degrade when there is high sparsity due to attempting to explain some of the signal by correlations that are not really there.  Results we incorporate in the revised version with the Criteo dataset demonstrate that the full matrix updates are also inferior in practice for such datasets.  With known collinear subsets of covariates one can use updates such as the multi-dimensional matrix update described in Appendix B.1 (now D.1). We extended the description of this method in the revision.  Finally, as shown in (Kakade, Ng. 2005, Shamir 2020), a distribution over the parameter is a means of performing Bayesian mixture on the unknown value of each parameter, such that the final posterior would tend to a point mass around the regret minimizer value.  Correlated covariates change the interim behavior early on, but at the horizon, even if covariates are correlated, the algorithm will converge to almost point masses.  In practice, this is not the case, however, as many covariates occur a limited number of times.
>
> - **Gaussian Approximation:** This is standard, and appears, e.g., in Kevin Murphy’s book and other literature. The loss due to approximating the Sigmoid by a Gaussian appears rather small relative to the approximation of the posterior as normal diagonal. We are re-incorporating some illustrations in an appendix, which we omitted from the text of the original submission, due to page limits, that show how close the approximation is to the Sigmoid.
>
> - **Complexity/speed:** Run times were proportional to the number of iterations. With the variational methods, even with 1000 examples per data point, regret fell short of the new approach, but the iterations were expectedly orders of magnitude slower, even when using highly optimized vector operations (as in the Eigen package).  Roughly, runtime increases by O(N) if we apply Monte Carlo on N samples.  We incorporated time benchmarks in the paper that clearly demonstrate the substantial speed improvements beyond what we had explained in the original text.  Benchmarks show that the Newton iterations do not noticeably slow down the first method, as they seem to converge rather fast, with a very small number of iterations (unlike the variational methods).  However, the second method, using further approximation (eq. (17) instead of (15) for updating mu), does not require Newton iterations at all, and is as good empirically.
>
> - **Probit Regression:** One can certainly use this approach with probit regression.  We added the equations for a similar algorithm with probit regression in an appendix.  However, a) logistic regression has been much more common in practice (possibly due to the popularity of gradient methods, where gradients are much simpler with logistic regression), and b) Probit would still require numerical methods, (which are, however, available in programming languages,) for evaluating the Gaussian CDF.

---

> > ### Author Response · Authors · 2020-11-25
> > **Response to AnonReviwer4 - part 2**
> >
> > - **Theoretical rigor:** We added clarifications on the Gaussian approximation of the Sigmoid. We believe that aggregate losses from this approximation are minor, and the actual loss comes from the approximations of the posterior. Regarding regret: while many papers studied practical Bayesian methods, we have not seen papers in the literature that study regret for practical Bayesian methods. Papers on Variational Bayes, expectation propagation, and other Bayesian methods propose heuristics to match the posterior, but do not consider regret at all. While theoretical papers (Kakade & Ng. 2005, Foster et. al. 2018, Shamir 2020) show the relationship between Bayesian methods and online regret, to the best of our knowledge, our paper is first to study regret for practical Bayesian methods. Proving regret bounds with the Gaussian and diagonal approximations can be a full COLT paper by itself. This paper focuses on showing empirically, as a proof of concept, that our proposed methodology actually works without the tedious complexity that Variational Bayes requires, and better than expectation propagation and other methods.  We believe that the empirical results demonstrate that.  We also believe that only synthetic data with known ground truth can demonstrate results on regret.  While we incorporated results on benchmark data, they cannot demonstrate the regret behavior, as there is no known optimal ground truth.
> >
> > - **Minor comments:** We corrected the reference oversight pointed out by the reviewer.

---

### Author Response · Authors · 2021-02-13
**Response from Authors to final decision**

We thank the reviewers and the area chairs for their work and efforts, and agree with the comments made by the area chair that (other) practical papers in the literature on this problem have no theoretical guarantees, and that all Bayesian methods depend on the choice of the prior.

We understand that the decision to reject the paper was eventually motivated by the lack of benchmark results.  However, we respectfully strongly disagree with this argument.  The paper presents an algorithm for stationary data.  In the lack of theoretical guarantees, the best way to demonstrate the merits of the method is by showing simulations on synthetic data, which eliminate other effects.  We do not claim to address non-stationarity, misspecification, or other phenomena, which exist in benchmark datasets, we claim we have a practical low-complexity algorithm for stationary data, which can be used as a building block for more sophisticated datasets.  Benchmark datasets are rich with other effects, which may hide the benefits of a building block type of algorithm.  We thus believe that the simulations we provided in the paper clearly demonstrate the advantages of the method provided for the problem that is addressed.

We separately comment on the post rebuttal comments by AnonReviewer3, which we believe are irrelevant to this work.

---

### Decision · Program_Chairs · 2021-01-07
**Final Decision**

**Decision:**

Reject

**Comment:**

This paper proposes a new approximate algorithm for Bayesian logistic regression in the online setting. The primary approximation involved in the algorithm is the use of a diagonal Gaussian approximation. (A probably more minor approximation is approximating the sigmoid with a Gaussian.) The main discussion focused on two issues: Firstly, there was some sentiment that the paper lacked theoretical guarantees. Second, there were concerns about the experimental results. I feel that it is not a serious flaw that the paper lacks a theoretical regret bound. Given the current state of algorithms for this problem practical algorithms remain very much of interest. However, the general sentiment of reviewers was that the experimental results were not as strong (or as numerous) as would be hoped. For an algorithm without a theoretical regret bound, I do agree that stronger empirical evidence would be expected. This was partially addressed in a revision but still I agree with the consensus that more extensive numerical evidence should be expected, and for that reason I am recommending rejection.

Finally, I'll mention some other issues that I view as not counting substantially against the paper. Firstly is the dependence on the prior. Here I am in agreement with the authors that this is an aspect shared by all Bayesian methods. This issue is a (valid) argument about the value of all Bayesian methods, but not one I think we will resolve here. Second, there were suggestions from the reviewers about improvements that could be made to the baseline methods. Here I don't feel that it's fair that we ask the authors to make novel improvements to other algorithms, unless those improvements are very "obvious".